# On Temperature Scaling and Conformal Prediction of Deep Classifiers

Lahav Dabah [1]   Tom Tirer [1]

## Abstract

In many classification applications, the prediction of a deep neural network (DNN) based classifier needs to be accompanied by some confidence indication. Two popular approaches for that aim are: 1) *Calibration*: modifies the classifier's softmax values such that the maximal value better estimates the correctness probability; and 2) *Conformal Prediction* (CP): produces a prediction set of candidate labels that contains the true label with a user-specified probability, guaranteeing marginal coverage but not, e.g., per class coverage. In practice, both types of indications are desirable, yet, so far the interplay between them has not been investigated. Focusing on the ubiquitous *Temperature Scaling* (TS) calibration, we start this paper with an extensive empirical study of its effect on prominent CP methods. We show that while TS calibration improves the class-conditional coverage of adaptive CP methods, surprisingly, it negatively affects their prediction set sizes. Motivated by this behavior, we explore the effect of TS on CP *beyond its calibration application* and reveal an intriguing trend under which it allows to trade prediction set size and conditional coverage of adaptive CP methods. Then, we establish a mathematical theory that explains the entire non-monotonic trend. Finally, based on our experiments and theory, we offer guidelines for practitioners to effectively combine adaptive CP with calibration, aligned with user-defined goals.

## 1. Introduction

Modern classification systems are typically based on deep neural networks (DNNs) (Krizhevsky et al., 2012; He et al., 2016; Huang et al., 2017). In many applications, it is necessary to quantify and convey the level of uncertainty associated with each prediction of the DNN. This is crucial in high-stakes scenarios where human lives are at risk, such as medical diagnoses (Miotto et al., 2018) and autonomous vehicle decision-making (Grigorescu et al., 2020).

In practice, DNN classification models typically generate a post-softmax vector, akin to a probability vector with non-negative entries that add up to one. One might, intuitively, be interested in using the value associated with the prediction as the confidence (Cosmides & Tooby, 1996). However, this value often deviates substantially from the actual correctness probability. This discrepancy, known as *miscalibration*, is prevalent in modern DNN classifiers, which frequently demonstrate overconfidence: the maximal softmax value surpasses the true correctness probability (Guo et al., 2017). To address this issue, post-processing *calibration* methods are employed to adjust the values of the softmax vector. In particular, Guo et al. (2017) demonstrated the usefulness of a simple Temperature Scaling (TS) procedure (a single parameter variant of Platt scaling (Platt et al., 1999)). Since then, TS calibration has gained massive popularity (Liang et al., 2018; Ji et al., 2019; Wang et al., 2021; Frenkel & Goldberger, 2021; Ding et al., 2021; Wei et al., 2022). Thus studying it is of high significance.

Another post-processing approach for uncertainty indication is Conformal Prediction (CP), which was originated in (Vovk et al., 1999; 2005) and has attracted much attention recently. CP algorithms are based on devising scores for all the classes per sample (based on the softmax values) that are used for producing a set of predictions instead of a single predicted class. These methods have theoretical guarantees for *marginal coverage*: given a user-specified probability, the produced set will contain the true label with this probability, assuming that the data samples are exchangeable (e.g., the samples are i.i.d.). Note that this property does not ensure *conditional coverage*, i.e., coverage of the true label with the specified probability when conditioning the data, e.g., to a specific class. Consequently, CP methods are usually compared by both their prediction set sizes and their conditional coverage performance.

Clearly, in critical applications, both calibration and CP are desirable, as they provide complementary types of information that can lead to a comprehensive decision. However, so far the interplay between them remains largely unexplored.

---

[1]Faculty of Engineering, Bar-Ilan University, Ramat Gan, Israel. Correspondence to: Lahav Dabah <lahavdabah@gmail.com>.

*Proceedings of the 42nd International Conference on Machine Learning*, Vancouver, Canada. PMLR 267, 2025. Copyright 2025 by the author(s).

Specifically, the works (Angelopoulos et al., 2021; Lu et al., 2022; Gibbs et al., 2023; Lu et al., 2023) apply initial *TS calibration* (rather than any other calibration method) before applying their CP methods. Yet, none of them investigates what is the effect of this procedure on the CP methods.

In this work, we study the effect of TS, arguably the most common calibration technique, on three prominent CP methods: Least Ambiguous set-valued Classifier (LAC) (Lei & Wasserman, 2014; Sadinle et al., 2019), Adaptive Prediction Sets (APS) (Romano et al., 2020), and Regularized Adaptive Prediction Sets (RAPS) (Angelopoulos et al., 2021). Note that LAC (aka THR), APS and RAPS are probably the three most popular CP methods for classification. Indeed, important papers in this field, such as (Angelopoulos et al., 2021) and (Stutz et al., 2022), do not consider any other CP method. Our discoveries on the effect of TS calibration in this paper have motivated us to explore the TS mechanism also beyond its calibration application.

Our contributions can be summarized as follows:

- We conduct an extensive empirical study on DNN classifiers that shows that an initial TS calibration affects CP methods differently. Specifically, we show that its effect is negligible for LAC, but intriguing for adaptive methods (APS and RAPS): while their class-conditional coverage is improved, surprisingly, their prediction set sizes typically become larger.

- Following these findings, we investigate the impact of TS on CP *for a wide range of temperatures, beyond the values used for calibration*. We reveal that by modifying the temperature, TS enables to trade the prediction set sizes and the class-conditional coverage performance for RAPS and APS. Moreover, metrics of these properties display a similar *non-monotonic* pattern across all models and datasets examined.

- We present a rigorous theoretical analysis of the impact of TS on the prediction set sizes of APS and RAPS, offering a comprehensive explanation for the complex non-monotonic patterns observed empirically.

- Based on our theoretically-backed findings, we propose practical guidelines to effectively combine adaptive CP methods with TS calibration, which allow users to control the prediction set sizes and conditional coverage trade-off.

## 2. Background and Related Work

Let us present the notations that are used in the paper, followed by some preliminaries on TS and CP. We consider a $C$-classes classification task of the data $(X, Y)$ distributed on $\mathbb{R}^d \times [C]$, where $[C] := \{1, \ldots, C\}$. The classification is tackled by a DNN that for each input sample

$\mathbf{x} \in \mathbb{R}^d$ produces a logits vector $\mathbf{z} = \mathbf{z}(\mathbf{x}) \in \mathbb{R}^C$ that is fed into a final softmax function $\boldsymbol{\sigma} : \mathbb{R}^C \to \mathbb{R}^C$, defined as $\sigma_i(\mathbf{z}) = \dfrac{\exp(z_i)}{\sum_{j=1}^{C} \exp(z_j)}$. Typically, the post-softmax vector $\hat{\boldsymbol{\pi}}(\mathbf{x}) = \boldsymbol{\sigma}(\mathbf{z}(\mathbf{x}))$ is being treated as an estimate of the class probabilities. The predicted class is given by $\hat{y}(\mathbf{x}) = \operatorname{argmax}_i \hat{\pi}_i(\mathbf{x})$.

### 2.1. Calibration and Temperature Scaling

The interpretation of $\hat{\boldsymbol{\pi}}(\mathbf{x})$ as an estimated class probabilities vector promotes treating $\hat{\pi}_{\hat{y}(\mathbf{x})}(\mathbf{x})$ as the probability that the predicted class $\hat{y}(\mathbf{x})$ is correct, also referred to as the model's *confidence*. However, it has been shown that DNNs are frequently overconfident — $\hat{\pi}_{\hat{y}(x)}(x)$ is larger than the true correctness probability (Guo et al., 2017). Formally, $\mathbb{P}\left(\hat{y}(X) = Y | \hat{\pi}_{\hat{y}(X)}(X) = p\right) < p$ with significant margin. Post-processing calibration techniques aim at reducing the aforementioned gap. They are based on optimizing certain transformations of the logits $\mathbf{z}(\cdot)$, yielding a probability vector $\tilde{\boldsymbol{\pi}}(\cdot)$ that minimizes an objective computed over a dedicated *calibration set* of labeled samples $\{\mathbf{x}_i, y_i\}_{i=1}^n$ (Platt et al., 1999; Zadrozny & Elkan, 2002; Naeini et al., 2015; Nixon et al., 2019). Two popular calibration objectives are the Negative Log-Likelihood (NLL) (Hastie et al., 2005) and the Expected Calibration Error (ECE) (Naeini et al., 2015), detailed in Appendix B.3.

Temperature Scaling (TS) (Guo et al., 2017) stands out as, arguably, the most common calibration approach, surpassing many others in achieving calibration with minimal computational complexity (Liang et al., 2018; Ji et al., 2019; Wang et al., 2021; Frenkel & Goldberger, 2021; Ding et al., 2021; Wei et al., 2022). It simply uses the transformation $\mathbf{z} \mapsto \mathbf{z}/T$ before applying the softmax, where $T > 0$ (the *temperature*) is a single scalar parameter that is set by minimizing NLL or ECE. *Importantly*, TS preserves the accuracy rate of the network (the ranking of the elements – and in particular, the index of the maximum – is unchanged).

Hereafter, we use the notation $\hat{\boldsymbol{\pi}}_T(\mathbf{x}) := \boldsymbol{\sigma}(\mathbf{z}(\mathbf{x})/T)$ to denote the output of the softmax when taking into account the temperature. Observe that $T = 1$ preserves the original probability vector. Let us denote by $T^*$ the temperature that is optimal for TS calibration. Since DNN classifiers are commonly overconfident, TS calibration typically yields some $T^* > 1$, which "softens" the original probability vector. Formally, TS with $T > 1$ raises the entropy of the softmax output (see Proposition A.5 in the appendix).

The *reliability diagram* (DeGroot & Fienberg, 1983; Niculescu-Mizil & Caruana, 2005) is a graphical depiction of a model before and after calibration. In Appendix B.3.2, we provide details on this procedure and display reliability diagrams for the dataset-model pairs examined in our study.

## 2.2. Conformal Prediction

Conformal Prediction (CP) is a methodology that is model-agnostic and distribution-free, designed for generating a *prediction set* of classes $\mathcal{C}_\alpha(X)$ for a given sample $X$, such that $Y \in \mathcal{C}_\alpha(X)$ with probability $1 - \alpha$ for a predefined $\alpha \in (0, 1)$, where $Y$ is the true class associated with $X$ (Vovk et al., 1999; 2005; Papadopoulos et al., 2002). The decision rule is based on a calibration set of labeled samples $\{\mathbf{x}_i, y_i\}_{i=1}^n$, which we hereafter refer to as the *CP set*, to avoid confusion with the set used for TS calibration. The only assumption in CP is that the random variables associated with the CP set and the test samples are exchangeable (e.g., the samples are i.i.d.).

Let us state the general process of conformal prediction given the CP set $\{\mathbf{x}_i, y_i\}_{i=1}^n$ and its deployment for a new (test) sample $x_{n+1}$ (for which $y_{n+1}$ is unknown), as presented in (Angelopoulos & Bates, 2021):

1. Define a heuristic score function $s(\mathbf{x}, y) \in \mathbb{R}$ based on some output of the model. A higher score should encode a lower level of agreement between $\mathbf{x}$ and $y$.

2. Compute $\hat{q}$ as the $\dfrac{\lceil (n+1)(1-\alpha) \rceil}{n}$ quantile of the scores $\{s(\mathbf{x}_1, y_1), \ldots, s(\mathbf{x}_n, y_n)\}$.

3. At deployment, use $\hat{q}$ to create prediction sets for new samples: $\mathcal{C}_\alpha(\mathbf{x}_{n+1}) = \{y : s(\mathbf{x}_{n+1}, y) \leq \hat{q}\}$.

CP methods possess the following coverage guarantee.

**Theorem 2.1** (Theorem 1 in (Angelopoulos & Bates, 2021))**.** *Suppose that $\{(X_i, Y_i)\}_{i=1}^n$ and $(X_{n+1}, Y_{n+1})$ are i.i.d., and define $\hat{q}$ as in step 2 above and $\mathcal{C}_\alpha(X_{n+1})$ as in step 3 above. Then the following holds:*

$$\mathbb{P}\left(Y_{n+1} \in \mathcal{C}_\alpha(X_{n+1})\right) \geq 1 - \alpha. \tag{1}$$

The proof of this result is based on (Vovk et al., 1999). A proof of an upper bound of $1 - \alpha + 1/(n+1)$ also exists. This property is called *marginal coverage* since the probability is taken over the entire distribution of $(X, Y)$.

Note that this property does not imply the much more stringent property of *conditional coverage*:

$$\mathbb{P}\left(Y_{n+1} \in \mathcal{C}_\alpha(X_{n+1}) | X_{n+1} = \mathbf{x}\right) \geq 1 - \alpha. \tag{2}$$

In fact, coverage for any value $\mathbf{x}$ of the random $X$ is impracticable (Vovk, 2012), (Foygel Barber et al., 2021), and a useful intuitive relaxation is to consider class-conditional coverage.

CP methods are usually compared by the size of their prediction sets and by their proximity to the conditional coverage property. Over time, various CP techniques with distinct objectives have been developed (Angelopoulos & Bates, 2021).

There have been also efforts to alleviate the exchangeability assumption (Tibshirani et al., 2019; Barber et al., 2023). In this paper we will focus on three prominent CP methods. Each of them devises a different score $s(\mathbf{x}, y)$ based on the output of the classifier's softmax $\hat{\boldsymbol{\pi}}(\cdot)$.

**Least Ambiguous Set-valued Classifier (LAC)** (Lei & Wasserman, 2014; Sadinle et al., 2019). In this method, $s(\mathbf{x}, y) = 1 - \hat{\pi}_y(\mathbf{x})$. Accordingly, given $\hat{q}_{LAC}$ associated with $\alpha$ through step 2, the prediction sets are formed as: $\mathcal{C}^{LAC}(\mathbf{x}) := \{y : \hat{\pi}_y(\mathbf{x}) \geq 1 - \hat{q}_{LAC}\}$. LAC tends to have small set sizes (under the strong assumption that $\hat{\boldsymbol{\pi}}(\mathbf{x})$ matches the posterior probability, it provably gives the smallest possible average set size). On the other hand, its conditional coverage is limited.

**Adaptive Prediction Sets (APS)** (Romano et al., 2020). The objective of this method is to improve the conditional coverage. Motivated by theory derived under the strong assumption that $\hat{\boldsymbol{\pi}}(\mathbf{x})$ matches the posterior probability, it uses $s(\mathbf{x}, y) = \sum_{i=1}^{L_y} \hat{\pi}_{(i)}(\mathbf{x})$, where $\hat{\pi}_{(i)}(\mathbf{x})$ denotes the $i$-th element in a descendingly sorted version of $\hat{\boldsymbol{\pi}}(\mathbf{x})$ and $L_y$ is the index that $y$ is permuted to after sorting. Following steps 2 and 3, yields $\hat{q}_{APS}$ and $\mathcal{C}^{APS}(\mathbf{x})$.

**Regularized Adaptive Prediction Sets (RAPS)** (Angelopoulos et al., 2021). A modification of APS that aims at improving its prediction set sizes by penalizing hard examples to reduce their effect on $\hat{q}$. With the same notation as APS, in RAPS we have $s(\mathbf{x}, y) = \sum_{i=1}^{L_y} \hat{\pi}_{(i)}(\mathbf{x}) + \lambda(L_y - k_{reg})_+$, where $\lambda, k_{reg} \geq 0$ are regularization hyperparameters and we use the notation $(\cdot)_+ := \max\{\cdot, 0\}$. Following steps 2 and 3, yields $\hat{q}_{RAPS}$ and $\mathcal{C}^{RAPS}(\mathbf{x})$.

Note that all these CP methods can be readily applied on $\boldsymbol{\pi}_{T^*}(\cdot)$ after TS calibration. This is done, in (Angelopoulos et al., 2021; Lu et al., 2022; Gibbs et al., 2023; Lu et al., 2023), where the authors stated that they applied TS calibration before examining the CP techniques. However, none of these works has examined how TS calibration impacts any CP method. All the more so, no prior work has experimented applying TS with a range of temperatures before employing CP methods.

After completing our work, we became aware of a concurrent work (Xi et al., 2024), which also studies the effect of TS on CP methods. Yet, their results are only a small subset of our results. They do not consider conditional coverage and explore only a limited range of temperatures, which masks the non-monotonic effect on the prediction set size of APS and RAPS (which they do not compare to LAC). In contrast, our paper provides a complete picture of the effect of TS across a wide range of temperatures on both the prediction set size and the class-conditional coverage of APS, RAPS, and LAC. This complete picture shows practitioners

that tuning the temperature for APS and RAPS introduces a trade-off effect, and we provide a practical way to control it. Lastly, note that several theoretical results on prediction sets and calibration have been reported in (Gupta et al., 2020). Yet, that work is limited to binary classification and does not provide an explanation for the fact that calibration affects CP methods differently and in a non-monotonic manner, as shown and analyzed in our paper.

## 3. The Effect of TS on CP Methods for DNN Classifiers

In this section, we empirically investigate the effect of TS on the performance of CP algorithms. Specifically, we consider different datasets and models, and start by reporting the mean prediction set size, marginal coverage, and class-conditional coverage of CP algorithms, with and without an initial TS calibration procedure. Then, we extend the empirical study to encompass a wide range of temperatures and discuss our findings.

### 3.1. Experimental setup

**Datasets.** We conducted our experiment on CIFAR-10, CIFAR-100 (Krizhevsky et al., 2009) and ImageNet (Deng et al., 2009) chosen for their diverse content and varying levels of difficulty.

**Models.** We utilized a diverse set of DNN classifiers, based on ResNets (He et al., 2016), DenseNets (Huang et al., 2017) and ViT (Dosovitskiy et al., 2021).

For CIFAR-10: ResNet34 and ResNet50. For CIFAR-100: ResNet50 and DenseNet121. For ImageNet: ResNet152, DenseNet121 and ViT-B/16. Details on the training of the models are provided in Appendix B.1.

**TS calibration.** For each dataset-model pair, we create a calibration set by randomly selecting 10% of the validation set. We obtain the calibration temperature $T^*$ by optimizing the ECE objective. The optimal temperatures when using the NLL objective are very similar, as displayed in Table 3 in the Appendix B.3.1. This justifies using ECE as the default for the experiments.

**CP Algorithms.** For each of the dataset-model pairs, we construct the "CP set" (used for computing the thresholds of CP methods) by randomly selecting $\{5\%, 10\%, 20\%\}$ of the validation set, while ensuring not to include in the CP set samples that are used in the TS calibration. The CP methods that we examine are LAC, APS, and RAPS, detailed in Section 2.2 (we use the randomized versions of APS and RAPS, as done in (Angelopoulos et al., 2021)). For each technique, we use $\alpha = 0.1$ and $\alpha = 0.05$, so the desired marginal coverage probability is 90% and 95%, as common in most CP literature (Romano et al., 2020; Angelopoulos

et al., 2021; Angelopoulos & Bates, 2021).

**Metrics.** We report metrics over the validation set, which we denote by $\{(\mathbf{x}_i^{(val)}, y_i^{(val)})\}_{i=1}^{N_{val}}$, comprising of the samples that were not included in the calibration set or CP set. The metrics are as follows.

● *Average set size* (AvgSize) – The mean prediction set size of the CP algorithm:

$$\text{AvgSize} = \frac{1}{N_{val}} \sum_{i=1}^{N_{val}} |\mathcal{C}(\mathbf{x}_i^{(val)})|.$$

● *Marginal coverage gap* (MarCovGap) – The deviation of the marginal coverage from the desired $1 - \alpha$:

$$\text{MarCovGap} =$$

$$\left| \frac{1}{N_{val}} \sum_{i=1}^{N_{val}} \mathbb{1}\{y_i^{(val)} \in \mathcal{C}(\mathbf{x}_i^{(val)})\} - (1 - \alpha) \right|.$$

● *Top-5% class-coverage gap* (TopCovGap) – The deviation from the desired $1 - \alpha$ coverage, averaged over the 5% of classes with the highest deviation:

$$\text{TopCovGap} =$$

$$\text{Top5}_{y \in [C]} \left| \frac{1}{|I_y|} \sum_{i \in I_y} \mathbb{1}\left\{ y_i^{(val)} \in \mathcal{C}(\mathbf{x}_i^{(val)}) \right\} - (1 - \alpha) \right|,$$

where $\text{Top5}$ is an operator that returns the mean of the 5% highest elements in the set and $I_y = \{i \in [N_{val}] : y_i^{(val)} = y\}$ is the indices of validation examples with label $y$. We use top-5% classes deviation due to the high variance in the maximal class deviation.

We also evaluate the *average class-coverage gap* (AvgCov-Gap) – a metric that measures the deviation from the target $1 - \alpha$ coverage, averaged across all classes. The experimental results for this metric closely mirror those observed with the TopCovGap metric. Thus, we defer its formal definition and experimental evaluation to Appendix B.5.3.

Note that for these metrics: *the lower the better*. Similar metrics have been used in (Ding et al., 2023; Angelopoulos et al., 2021). All the reported results, per metric, are the median-of-means along 100 trials where we randomly select the calibration/CP sets, similarly to (Angelopoulos et al., 2021).

### 3.2. The effect of TS calibration on CP methods

For each of the dataset-model pairs we compute the afore-mentioned metrics with and without an initial TS calibra-tion procedure. In Table 1, we report the the calibration temperature $T^*$, the accuracy (not affected by TS), and the median-of-means of the prediction set sizes metric, AvgSize.

*Table 1.* **Prediction Set Size.** AvgSize metric along with $T^*$ and accuracy for dataset-model pairs using LAC, APS, and RAPS algorithms with $\alpha = 0.1$, CP set size 10%, pre- and post-TS calibration.

| | | Accuracy(%) | | AvgSize | | | AvgSize after TS | | |
| Dataset-Model | $T^*$ | Top-1 | Top-5 | LAC | APS | RAPS | LAC | APS | RAPS |
|---|---|---|---|---|---|---|---|---|---|
| ImageNet, ResNet152 | 1.227 | 78.3 | 94.0 | 1.95 | 6.34 | 2.71 | 1.92 | 11.11 | 4.30 |
| ImageNet, DenseNet121 | 1.024 | 74.4 | 91.9 | 2.73 | 9.60 | 4.70 | 2.76 | 11.32 | 4.88 |
| ImageNet, ViT-B/16 | 1.180 | 83.9 | 97.0 | 2.22 | 10.10 | 1.93 | 2.23 | 19.27 | 2.34 |
| CIFAR-100, ResNet50 | 1.524 | 80.9 | 95.4 | 1.62 | 5.31 | 2.88 | 1.57 | 9.14 | 4.96 |
| CIFAR-100, DenseNet121 | 1.469 | 76.1 | 93.5 | 2.13 | 4.26 | 2.98 | 2.06 | 6.51 | 4.27 |
| CIFAR-10, ResNet50 | 1.761 | 94.6 | 99.7 | 0.91 | 1.04 | 0.98 | 0.91 | 1.13 | 1.05 |
| CIFAR-10, ResNet34 | 1.802 | 95.3 | 99.8 | 0.91 | 1.03 | 0.94 | 0.93 | 1.11 | 1.05 |

*Table 2.* **Coverage Metrics.** MarCovGap and TopCovGap metrics for dataset-model pairs using LAC, APS, and RAPS algorithms with $\alpha = 0.1$, CP set size 10%, pre- and post-TS calibration.

| | MarCovGap(%) | | | MarCovGap TS(%) | | | TopCovGap(%) | | | TopCovGap TS(%) | | |
| Dataset-Model | LAC | APS | RAPS | LAC | APS | RAPS | LAC | APS | RAPS | LAC | APS | RAPS |
|---|---|---|---|---|---|---|---|---|---|---|---|---|
| ImageNet, ResNet152 | 0.1 | 0 | 0 | 0 | 0 | 0 | 23.1 | 16.0 | 17.6 | 23.9 | 13.8 | 15.2 |
| ImageNet, DenseNet121 | 0 | 0.1 | 0 | 0.1 | 0 | 0 | 24.9 | 15.7 | 18.0 | 25.2 | 14.9 | 17.6 |
| ImageNet, ViT-B/16 | 0 | 0 | 0 | 0.1 | 0.1 | 0 | 24.8 | 14.2 | 14.7 | 24.9 | 12.2 | 12.5 |
| CIFAR-100, ResNet50 | 0.1 | 0 | 0 | 0 | 0.1 | 0 | 13.9 | 12.6 | 11.7 | 12.9 | 9.0 | 7.9 |
| CIFAR-100, DenseNet121 | 0 | 0 | 0 | 0 | 0 | 0.1 | 11.5 | 9.5 | 9.7 | 12.2 | 7.8 | 8.0 |
| CIFAR-10, ResNet50 | 0 | 0 | 0 | 0 | 0.1 | 0 | 11.1 | 5.0 | 4.8 | 11.2 | 2.4 | 2.6 |
| CIFAR-10, ResNet34 | 0 | 0 | 0.1 | 0 | 0 | 0 | 9.5 | 3.0 | 2.8 | 9.1 | 2.2 | 2.2 |

In Table 2, we report the median-of-means of the marginal and conditional coverage metrics, MarCovGap and TopCov-Gap. In both tables, the specified coverage probability is 90% ($\alpha = 0.1$), and we use 10% of the samples for the CP set and 10% of the samples for the calibration set. Due to space limitation, the results for coverage probability of 95% ($\alpha = 0.05$) and sizes $\{5\%, 20\%\}$ of the CP sets are deferred to Appendix B.4. The insights gained from Tables 1 and 2 hold also for the deferred results.

Examining the results, we first see that the TS calibration temperatures, $T^*$, in Table 1 are greater than 1, indicating that the models exhibit overconfidence. The reliability diagrams before and after TS calibration are presented in Appendix B.3.2.

By examining MarCovGap in Table 2, we see that all CP methods maintain marginal coverage both with and without the initial TS procedure (the gap is at most 0.1%, i.e., 0.001). That is, TS calibration does not affect this property, which is consistent with CP theoretical guarantees (Theorem 2.1).

As for the conditional coverage, as indicated by the TopCov-Gap metric in Table 2, there is no distinct trend observed for the LAC method. On the other hand, in the adaptive CP methods, APS and RAPS, there is a noticeable improvement (TopCovGap decreases), especially when $T^*$ is high. We turn to examine Table 1, which reports the effect of TS calibration on the prediction set size of the CP methods. First, we see that for the CIFAR-10, where the models' accuracy is very high, the effect on AvgSize is minor. Second, we see that the effect on LAC is negligible also for other

datasets. (Note that while LAC has lower AvgSize than APS and RAPS, its conditional coverage is worse, as shown by TopCovGap in Table 2). Third, perhaps the most thought provoking observation, for APS and RAPS the TS calibration procedure has led to *increase* in the mean prediction set size. Especially, when the value of the optimal temperature $T^*$ is high. For instance, for ResNet50 on CIFAR-100, TS calibration increases AvgSize of APS from 5.31 to 9.14. This behavior is quite surprising.

In Appendix B.4.1, we verify that the increase in the mean set size for APS and RAPS is not caused by a small number of extreme outliers. We observe that for many samples in the validation set the prediction set size is increased due to TS calibration. Only for a small minority of the samples the prediction set decreases. Yet, this teaches us that we cannot make a universal (uniform) statement about the impact of TS on the set size of arbitrary sample, but rather consider a typical/average case.

### 3.3. TS beyond calibration

The intriguing observations regarding TS calibration — especially, *both positively and negatively affecting different aspects of APS and RAPS* — prompt us to explore the effects of TS on CP, beyond calibration. In Figure 1 we present the average prediction set size (AvgSize), the class-conditional coverage metric (TopCovGap), and the threshold value of the CP methods ($\hat{q}$) for temperatures ranging from 0.5 to 5 with an increment of 0.1. The reported results are the median-of-means for 100 trials. We present here three diverse dataset-model pairs, and defer the others to

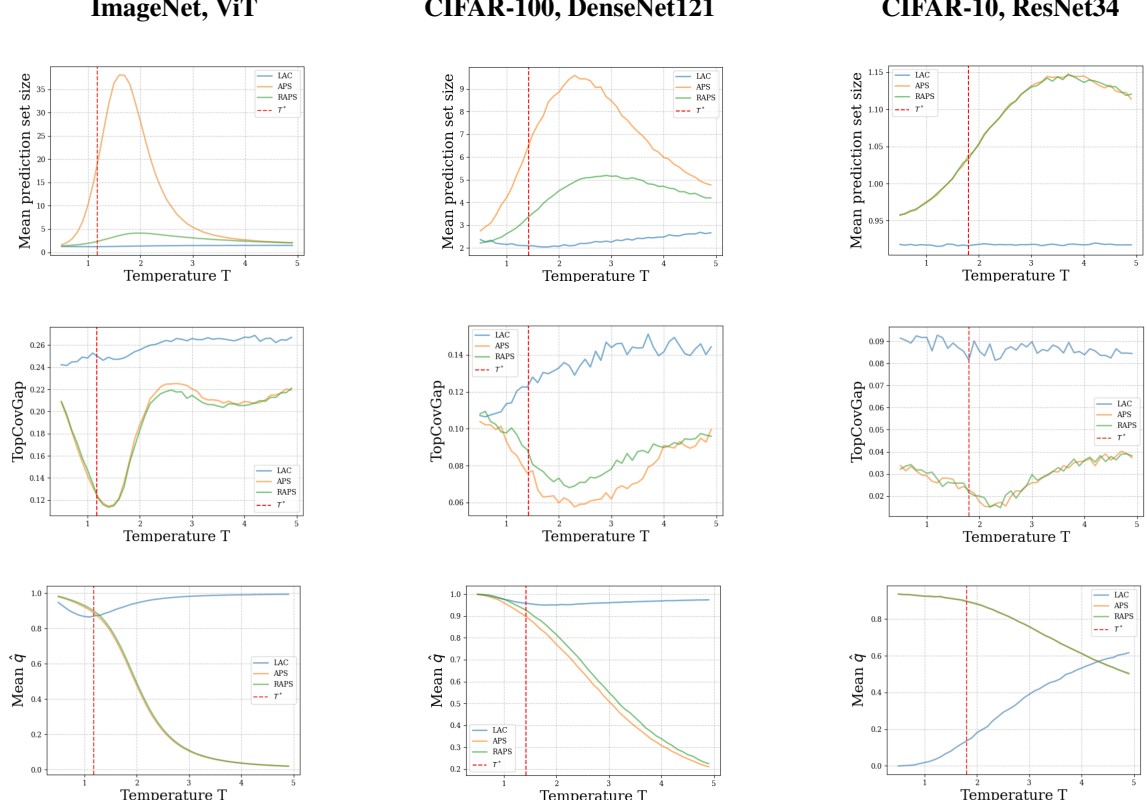

**ImageNet, ViT**     **CIFAR-100, DenseNet121**     **CIFAR-10, ResNet34**

*Figure 1.* AvgSize (top), TopCovGap (middle), and mean threshold $\hat{q}$ (bottom) for LAC, APS and RAPS with $\alpha = 0.1$ versus the temperature $T$. The vertical line marks $T^*$ obtained by calibration.

Appendix B.5 due to space limitation. The appendix also includes different settings, such as various calibration set sizes and coverage levels, detailed in Section 3.1.

Figure 1 displays interesting *non-monotonic* trends of Avg-Size and TopCovGap for APS and RAPS. Across all dataset-model combinations, these metrics exhibit similar patterns: AvgSize (top row) increases until reaching a peak, and then starts declining; TopCovGap (middle row) decreases until reaching a minimum, then reverses and starts increasing. The threshold value $\hat{q}$ (bottom row), on the other hand, decreases monotonically for APS and RAPS. For LAC no clear pattern is evident.

Let $T_c$ denote the "critical" temperature at which AvgSize reaches its peak (not to be confused with the optimal calibration temperature, $T^*$). Restricting our view to $T < T_c$, we see generalization of the observations of the TS calibration experiments (which compare $T = 1$ and $T = T^*$). Specifically, for the APS and RAPS algorithms, as $T$ increases AvgSize increases while TopCovGap decreases. This reveals that in this range of $T$, there is a *trade-off* between the two crucial properties of APS and RAPS – prediction set sizes and conditional coverage – which can be controlled by increasing/decreasing $T$. In fact, overall, the

trade-off remains also beyond the maximum/minimum of AvgSize/TopCovGap. Nevertheless, the existence of non-monotonic trends with extreme points is surprising and not intuitive.

## 4. Theoretical analysis

In this section, we provide mathematical reasoning for empirical observations regarding the effect of TS on the prediction set size of APS and RAPS presented in Section 3. Theoretically analyzing our other findings, such as the effect of TS on the conditional coverage, are left for future research. *All the proofs are provided in Appendix A.*

### 4.1. APS / RAPS threshold decreases as $T$ increases

In Section 3.3, we observe that increasing the temperature monotonically reduces the threshold of APS and RAPS. Let us prove this theoretically. Later, we will use this result in our theory on the effect of TS on the prediction set size.

Let $\mathbf{z} = \mathbf{z}(\mathbf{x})$ be the logits vector of a sample $\mathbf{x}$. Let $\hat{\boldsymbol{\pi}}_T = \boldsymbol{\sigma}(\mathbf{z}/T)$ be the softmax vector after TS with temperature $T$. We denote by $s_T(\mathbf{x}, y)$ the score of the APS method when

applied on $\hat{\boldsymbol{\pi}}_T$. Namely, $s_T(\mathbf{x}, y) = \sum_{i=1}^{L_y} \hat{\pi}_{T,(i)}$, where $\hat{\pi}_{T,(i)}$ denotes the $i$-th element in a descendingly sorted version of $\hat{\boldsymbol{\pi}}_T$ and $L_y$ is the index that $y$ is permuted to after sorting. Recall that the RAPS algorithm is based on the same score with an additional regularization term that is not affected by TS.

The following theorem states that a cumulative sum of a sorted softmax vector, analogous to the APS score, decreases as the temperature $T$ increases.

**Theorem 4.1.** *Let $\mathbf{z} \in \mathbb{R}^C$ be a sorted logits vector, i.e., $z_1 \geq z_2 \geq \ldots \geq z_C$, and let $L \in [C]$. Let $\hat{\boldsymbol{\pi}}_T = \boldsymbol{\sigma}(\mathbf{z}/T)$ and $\hat{\boldsymbol{\pi}}_{\tilde{T}} = \boldsymbol{\sigma}(\mathbf{z}/\tilde{T})$ with $T > \tilde{T} > 0$. Then, we have $\sum_{j=1}^{L} \pi_{\tilde{T},j} \geq \sum_{j=1}^{L} \pi_{T,j}$. The inequality is strict, unless $L = C$ or $z_1 = \ldots = z_C$.*

Note that Theorem 4.1 is universal: it holds for any sorted logits vector. Denote the threshold obtained by applying the CP method after TS by $\hat{q}_T$. Based on the universality of the theorem, we can establish that increasing the temperature $T$ decreases $\hat{q}_T$ for APS and RAPS.

**Corollary 4.2.** *The threshold value $\hat{q}_T$ of APS and RAPS decreases monotonically as $T$ increases.*

### 4.2. The effect of TS on prediction set sizes of APS

In Section 3.3, we observe a consistent dependency on the temperature parameter $T$ across all models and datasets: the mean prediction set size of APS and RAPS switches from increasing to decreasing as $T$ passes some value $T_c$. In this section, we provide a theoretical analysis to elucidate this behavior. For simplification we focus on APS.

Let $\hat{q}_T$ and $\hat{q}$ denote the thresholds of APS when applied with and without TS, respectively. For a new test sample $\mathbf{x}$ with logits vector $\mathbf{z}$ that is *sorted in descending order*, $\hat{\boldsymbol{\pi}}_T = \boldsymbol{\sigma}(\mathbf{z}/T)$ and $\hat{\boldsymbol{\pi}} = \boldsymbol{\sigma}(\mathbf{z})$ are the softmax outputs with and without TS, which are *sorted as well*. We denote $L_T = \min\{\ell : \sum_{i=1}^{\ell} \hat{\pi}_{T,i} \geq \hat{q}_T\}$, $L = \min\{\ell : \sum_{i=1}^{\ell} \hat{\pi}_i \geq \hat{q}\}$ as the prediction set sizes for this sample according to APS with and without TS, respectively.

We aim to investigate the conditions under which the events $L_T \geq L$ and $L_T \leq L$ occur. Since analyzing these events directly seems to be challenging, we leverage the following proposition that establishes alternative events that are sufficient conditions.

**Proposition 4.3.** *Let $\mathbf{z} \in \mathbb{R}^C$ such that $z_1 \geq z_2 \geq \cdots \geq z_C$, $\hat{\boldsymbol{\pi}}_T = \boldsymbol{\sigma}(\mathbf{z}/T)$ and $\hat{\boldsymbol{\pi}} = \boldsymbol{\sigma}(\mathbf{z})$. Let $\hat{q}, \hat{q}_T \in (0, 1]$ such that if $T > 1$ then $\hat{q} \geq \hat{q}_T$ and if $0 < T < 1$ then $\hat{q} \leq \hat{q}_T$. Let $L = \min\{\ell : \sum_{i=1}^{\ell} \hat{\pi}_i \geq \hat{q}\}$, $L_T = \min\{\ell :$*

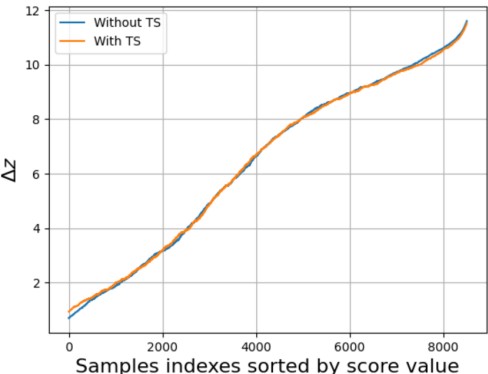

*Figure 2.* $\Delta z$ for each sample sorted by APS score value (from low to high), with and without TS calibration applied before sorting the scores, for CIFAR100-ResNet50.

$\sum_{i=1}^{\ell} \hat{\pi}_{T,i} \geq \hat{q}_T\}$. *The following holds:*

$$
\begin{cases}
\text{If} \quad 0 < T < 1 \text{ and } L_T > 1 \quad \text{then}: \\
\quad \sum_{i=1}^{L_T-1} \hat{\pi}_i - \sum_{i=1}^{L_T-1} \hat{\pi}_{T,i} \leq \hat{q} - \hat{q}_T \implies L \geq L_T \\
\text{If} \quad T > 1 \text{ and } L > 1 \quad \text{then}: \\
\quad \sum_{i=1}^{L-1} \hat{\pi}_i - \sum_{i=1}^{L-1} \hat{\pi}_{T,i} \geq \hat{q} - \hat{q}_T \implies L \leq L_T
\end{cases}
\tag{3}
$$

Note that our definitions of $L$ and $L_T$ imply that their minimal value is 1. Thus, in the case of $0 < T < 1$ and $L_T = 1$, we trivially have $L \geq L_T$. Similarly, in the case of $T > 1$ and $L = 1$, we trivially have $L \leq L_T$.

The right-hand side of the new inequalities pertains to the difference between the threshold values before and after applying TS. Analyzing this term requires understanding the properties of the quantile sample of APS.

Let $\mathbf{z}^q$ denote the logits vector of the sample associated with APS threshold (without applying TS), which we dub "the quantile sample". The CP theory builds on the score of the quantile sample being larger than $(1 - \alpha)\%$ of the scores of other samples with high probability.

As illustrated in Figure 2, there is a strong correlation between the score value and the difference $\Delta z := z_{(1)} - z_{(2)}$. This implies that, for typical values of $\alpha$ (e.g., 0.1), the quantile sample exhibits a highly dominant first entry in its logits vector $\mathbf{z}^q$.

Notably, even if the samples are sorted according to their scores *after applying TS*, this strong correlation still holds. Specifically, denoting by $\mathbf{z}_T^q$ the logits vector of the quantile sample when TS is applied, we observe in Figure 2 that it still exhibits a highly dominant first entry, with a $\Delta z$ value similar to that of $\mathbf{z}^q$. Recall that the softmax operation

applied to the logits vectors is invariant to constant shifts and further amplifies the dominance of larger entries ($\pi_i \propto \exp(z_i)$). Thus, having $z_{T,(1)}^q - z_{T,(2)}^q \approx z_{(1)}^q - z_{(2)}^q \gg 1$, practically implies that $\mathbf{z}_T^q \approx \mathbf{z}^q$ up to an additive constant. In Appendix B.6, we demonstrate this similarity across additional dataset–model pairs and temperature settings. Therefore, it is reasonable to make the technical assumption that both $\hat{q}$ and $\hat{q}_T$ correspond to the same sample in the CP set, denoted here by the sorted logits vector $\mathbf{z}^q$.

For the rest of the analysis, we define the "gap function":

$$
\begin{aligned}
g(\mathbf{z}; T, M) &:= \sum_{i=1}^{M} \sigma_i(\mathbf{z}) - \sum_{i=1}^{M} \sigma_i(\mathbf{z}/T) \\
&= \frac{\sum_{i=1}^{M} \exp(z_i)}{\sum_{j=1}^{C} \exp(z_j)} - \frac{\sum_{i=1}^{M} \exp(z_i/T)}{\sum_{j=1}^{C} \exp(z_j/T)} \quad (4)
\end{aligned}
$$

where $\mathbf{z}$ is a logits vector sorted in descending order.

With our assumption that $\hat{q}$ and $\hat{q}_T$ are associated with the same quantile sample $\mathbf{z}^q$, we have that $\hat{q} - \hat{q}_T$ in (3) can be written as $g(\mathbf{z}^q; T, L^q)$, where $L^q$ denotes the rank of the true label of $\mathbf{z}^q$.

In Proposition A.6 in Appendix A.1, we prove that $\forall M \in [C]$, $|g(\mathbf{z}^q, T, L^q) - g(\mathbf{z}^q, T, M)|$ decays exponentially with $\Delta z$. Due to the large $\Delta z$ of the quantile sample illustrated in Figure 2, we approximate $g(\mathbf{z}^q, T, L^q) \approx g(\mathbf{z}^q, T, M)$. Substituting $M = L_T - 1$ or $M = L - 1$, depending on the branch in (3), this justifies studying the following events:

$$
\begin{cases}
g(\mathbf{z}; T, M) \leq g(\mathbf{z}^q; T, M) & \text{if } 0 < T < 1 \\
g(\mathbf{z}; T, M) \geq g(\mathbf{z}^q; T, M) & \text{if } T > 1
\end{cases} \quad (5)
$$

Returning to consider $\mathbf{z}$ as the sorted logits vector of a test sample, typically it has lower score than the quantile sample $\mathbf{z}^q$, and thus as illustrated in Figure 2 also lower $\Delta z$. Consequently, it is intuitive to associate an increase in the score with an increase in the first entry of the sorted logits vector, $z_1$. We now present our key theorem that establishes a connection between the difference $\Delta z$ and the sign of $\nabla_{z_1} g(\mathbf{z}; T, M)$, depending on $T$ and on the following "bound function":

$$
b(T) := \begin{cases}
\text{If } \;\; 0 < T < 1: \\
\max \left\{ \dfrac{T}{T-1} \ln\left(\dfrac{T}{4}\right), \dfrac{T}{T+1} \ln\left(\dfrac{4(C-1)^2}{T}\right) \right\} \\
\text{If } \;\; T > 1: \\
\max \left\{ \dfrac{T}{T-1} \ln(4T), \dfrac{T}{T+1} \ln(4T(C-1)^2) \right\}
\end{cases}
$$
$$(6)$$

**Theorem 4.4.** *Let $\mathbf{z} \in \mathbb{R}^C$ such that $z_1 \geq z_2 \geq \cdots \geq z_C$ and denote $\Delta z = z_1 - z_2$. Then, the following holds:*

$$
\begin{cases}
\text{If } \;\; 0 < T < 1: & \Delta z > b(T) \implies \nabla_{z_1} g(\mathbf{z}; T, M) > 0 \\
\text{If } \;\; T > 1: & \Delta z > b(T) \implies \nabla_{z_1} g(\mathbf{z}; T, M) < 0
\end{cases}
$$
$$(7)$$

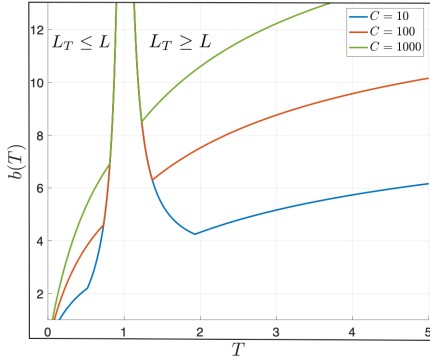

*Figure 3.* $b(T)$ of Thm. 4.4, for $C = \{10, 100, 1000\}$.

Let us consider the case of TS with $T > 1$. The theorem establishes that for a sample with a sorted logits vector $\mathbf{z}$ satisfying $\Delta z > b_{T>1}(T)$, we have that $g(\mathbf{z})$ decreases monotonically as $z_1$ increases (indeed, when $z_1$ increases then $\Delta z$ increases, and thus the inequality $\Delta z > b_{T>1}(T)$ remains satisfied). Since $\mathbf{z}^q$ has larger dominant entry than typical $\mathbf{z}$, this implies that $g(\mathbf{z}; T, M) > g(\mathbf{z}^q; T, M)$. Thus, under our technical assumption that $\hat{q}$ and $\hat{q}_T$ correspond to the same sample, we can apply Proposition 4.3 and get $L_T \geq L$ — a larger prediction set of APS. Conversely, by the same logic, the effect of TS with $0 < T < 1$ on a sample with a sorted logits vector $\mathbf{z}$ satisfying $\Delta z > b_{T<1}(T)$, is a smaller prediction set of APS.

We now show that the bounds in Theorem 4.4 do not require unreasonable values of $\Delta z$ and $T$. In particular, the theorem explains the phenomenon for a "typical" $\mathbf{z}$ — e.g., the sample with the median score in Figure 2 — which in turn explains why we see the increase/decrease of the *mean* prediction set. Indeed, for this sample, according to Figure 2 we have $\Delta z \approx 8$. For $C = 100$ (as in this CIFAR-100 experiment), the bounds in the theorem are complied by this sample for the temperature ranges $0 < T < 0.83$ and $1.25 < T < 2.33$. Since the calibrated temperature in this CIFAR100-ResNet50 experiment is $T^* = 1.524$, which falls in this range, we rigorously proved increased prediction set for the median sample after TS calibration. Interestingly, the broad temperature range that is covered by our theory indicates that our bounds are sufficiently tight to establish additional fine-grained insights.

**Implications of the bounds.** To demonstrate the significance of our theory, we analyze the bounds $b(T)$ established in Theorem 4.4. We leverage our theory to explain the entire non-monotonic trend in the empirical results on the effect of TS on the mean prediction set size of APS (and the closely related RAPS), showed in Figure 1.

In Figure 3 we present the bound as a function of $T$ for $C = \{10, 100, 1000\}$. According to the analysis, samples whose

$\Delta z$ is above the bound are those for which the TS operation yields a larger (resp. smaller) prediction set size when $T > 1$ (resp. $T < 1$). We denote by $\tilde{T}_c$ the temperature value at which the bound attains its minimum value for $T > 1$. This value can be computed (numerically) by the intersection of the functions in the $\max$ operation at the $T > 1$ branch in (6). Note that it is affected by the number of classes $C$. Inspecting Figure 3 we gain the following insights.

- For $0 < T < 1$: The bound increases as a function of $T$. Thus, as $T$ decreases, a greater proportion of samples satisfy the bound, which is aligned with a reduction in the mean prediction set sizes.

- For $1 < T < \tilde{T}_c$: The bound decreases as $T$ increases, indicating that more samples comply with the bound (have $L_T \geq L$), which is aligned with an increase in the mean prediction set sizes.

- At $T = \tilde{T}_c$: The bound attains a minimum, corresponding to the maximum number of samples satisfying the bound, which is aligned with the largest mean prediction set size.

- For $T > \tilde{T}_c$: The bound again increases, indicating that as $T$ continues to rise, fewer samples satisfy the bound, which is aligned with a decrease in the prediction set sizes.

We see that $\tilde{T}_c$ describes the temperature at which the trend of the prediction set size shifts from increasing to decreasing as $T$ increases. Our theory shows that $\tilde{T}_c$ shifts to lower temperatures as $C$ increases. A similar trend is observed in the empirical results shown in Figure 1.

## 5. Guidelines for Practitioners

Based on our theoretically-backed findings, we propose a guideline, depicted in Fig. 4, for practitioners that wish to use adaptive CP methods (e.g., due to their better conditional coverage). Specifically, we suggest to use TS with two different temperature parameters on separate branches: $T^*$ that is optimized for TS calibration, and $\hat{T}$ that allows trading the prediction set sizes and conditional coverage properties of APS/RAPS to better fit the task's requirements. Our experiments and theory show that $\hat{T}$ should be scanned up to a value $T_c$.

Note that one does not know in advance what values of the metrics are obtained per value of $\hat{T}$. However, since we propose to separate the calibration and the CP procedure, the calibration set can also be used to evaluate the CP algorithms *without dangering exchangeability*. Indeed, in Appendix C, we demonstrate how using a small amount of calibration data we can approximate the curves of AvgSize and TopCovGap vs. $T$ that appear in Figure 1 (which were generated using the entire validation set that is not accessible to the user in practice). According to the approximate

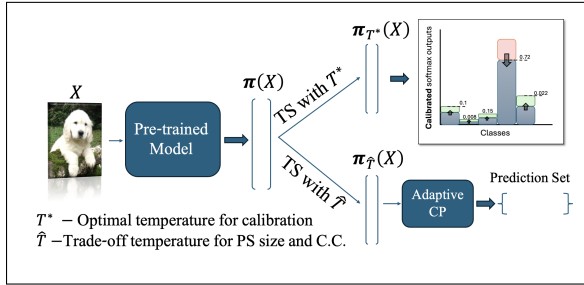

*Figure 4.* Guideline for using TS calibration and adaptive CP.

trends, the user can choose $\hat{T}$ that best fit their requirements. Furthermore, note that the procedure required to produce approximated curves of metrics vs. $T$ is done *offline* during the calibration phase and its runtime is on the order of minutes.

In Appendix D, we further show the practical significance of our findings and guidelines. Specifically, for users that prioritize class-conditional coverage, we show that applying TS with temperature of the minimum in the approximated TopCovGap curve (created using only the calibration set as explained in Appendix C) followed by RAPS outperforms Mondrian CP (Vovk, 2012), which is tailored to classwise CP, in both TopCovGap and AvgSize metrics. Code for reproducing our experiments and applying our guidelines is available at https://github.com/lahavdabah/TS4CP.

## 6. Conclusion

In this work, we studied the effect of the widely used TS calibration on the performance of CP techniques. These popular complementary approaches are useful for assessing the reliability of classifiers, in particular those that are based DNNs. Yet, their interplay has not been examined so far. We conducted an extensive empirical study on the effect of TS, even beyond its calibration application, on prominent CP methods. Among our findings, we discovered that TS enables trading prediction set size and class-conditional coverage performance of adaptive CP methods (APS and RAPS) through a non-monotonic pattern, which is similar across all models and datasets examined. We presented a theoretical analysis on the effect of TS on the prediction set sizes of APS and RAPS, which offers a comprehensive explanation for this pattern. Finally, based on our findings, we provided practical guidelines for combining APS and RAPS with calibration while adjusting them via a dedicated TS mechanism to better fit specific requirements.

## Acknowledgment

The work was supported by the Israel Science Foundation (No. 1940/23) and MOST (No. 0007091) grants.

## Impact Statement

Our work focuses on investigating the interplay between approaches aiming at making modern classifiers reliable. Our finding can guide practitioners to use these approaches better. Improved reliability estimation can significantly influence and guide society toward a better future.

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

# A. Proofs.

**Theorem A.1.** *Let* $\mathbf{z} \in \mathbb{R}^C$ *be a sorted logits vector, i.e.,* $z_1 \geq z_2 \geq \ldots \geq z_C$, *and let* $L \in [C]$. *Let* $\hat{\boldsymbol{\pi}}_T = \boldsymbol{\sigma}(\mathbf{z}/T)$ *and* $\hat{\boldsymbol{\pi}}_{\tilde{T}} = \boldsymbol{\sigma}(\mathbf{z}/\tilde{T})$ *with* $T > \tilde{T} > 0$. *Then, we have* $\sum_{j=1}^{L} \pi_{\tilde{T},j} \geq \sum_{j=1}^{L} \pi_{T,j}$. *The inequality is strict, unless* $L = C$ *or* $z_1 = \ldots = z_C$.

Before we turn to prove the theorem, let us prove an auxiliary lemma.

**Lemma.** *Let* $z_i, z_j \in \mathbb{R}$ *such that* $z_i \geq z_j$ *and let* $T \geq \tilde{T} \geq 0$. *Then, the following holds*

$$\exp(z_i/\tilde{T}) \cdot \exp(z_j/T) \geq \exp(z_i/T) \cdot \exp(z_j/\tilde{T}).$$

*The inequality is strict, unless* $T = \tilde{T}$ *or* $z_i = z_j$.

*Proof.* Given that $z_i - z_j \geq 0$ and $T \geq \tilde{T} \geq 0$, it follows that:

$$(z_i - z_j)/\tilde{T} \geq (z_i - z_j)/T.$$

Since the exponential function is monotonically increasing, applying it to both sides of the inequality yields:

$$\exp((z_i - z_j)/\tilde{T}) \geq \exp((z_i - z_j)/T).$$

Multiplying both sides by $\exp(z_j/T) \exp(z_j/\tilde{T})$, we obtain:

$$\exp(z_i/\tilde{T}) \cdot \exp(z_j/T) \geq \exp(z_i/T) \cdot \exp(z_j/\tilde{T}),$$

which is the desired result. The strict monotonicity of the exponential function implies that equality holds only if $(z_i - z_j)/\tilde{T} = (z_i - z_j)/T$, i.e., if $T = \tilde{T}$ or $z_i = z_j$.

$\square$

*Proof.* **Back to the proof of the theorem.**

Let $I = \{1, 2, \ldots, L\}$ and $J = \{L+1, L+2, \ldots, C\}$. Because $\mathbf{z}$ is sorted, $\forall i \in I, j \in J$ we have $z_i > z_j$. Therefore, according to the auxiliary lemma in Theorem A.1: $\exp(z_i/\tilde{T}) \cdot \exp(z_j/T) \geq \exp(z_i/T) \cdot \exp(z_j/\tilde{T})$ for any combination of $i \in I, j \in J$. Consequently, summing this inequality over all $i \in I, j \in J$, we get

$$\sum_{i=1}^{L} \sum_{j=L+1}^{C} \exp(z_i/\tilde{T}) \cdot \exp(z_j/T) \geq \sum_{i=1}^{L} \sum_{j=L+1}^{C} \exp(z_i/T) \cdot \exp(z_j/\tilde{T})$$

$$\Updownarrow$$

$$\sum_{i=1}^{L} \exp(z_i/\tilde{T}) \cdot \sum_{j=L+1}^{C} \exp(z_j/T) \geq \sum_{i=1}^{L} \exp(z_i/T) \cdot \sum_{j=L+1}^{C} \exp(z_j/\tilde{T}).$$

In the last line, we separated the summation of the indexes $i \in I, j \in J$.

Adding $\sum_{i=1}^{L} \exp(z_i/\tilde{T}) \cdot \sum_{j=1}^{L} \exp(z_j/T)$ to both sides, we get

$$\sum_{i=1}^{L} \exp(z_i/\tilde{T}) \left[ \sum_{i=1}^{L} \exp(z_i/T) + \sum_{j=L+1}^{C} \exp(z_j/T) \right] \geq \sum_{j=1}^{L} \exp(z_j/T) \left[ \sum_{i=1}^{L} \exp(z_i/\tilde{T}) + \sum_{j=L+1}^{C} \exp(z_j/\tilde{T}) \right]$$

$$\Updownarrow$$

$$\sum_{i=1}^{L} \exp(z_i/\tilde{T}) \sum_{j=1}^{C} \exp(z_j/T) \geq \sum_{i=1}^{L} \exp(z_i/T) \sum_{j=1}^{C} \exp(z_j/\tilde{T})$$

$$\Updownarrow$$

$$\frac{\sum_{i=1}^{L} \exp(z_i/\tilde{T})}{\sum_{j=1}^{C} \exp(z_j/\tilde{T})} \geq \frac{\sum_{i=1}^{L} \exp(z_i/T)}{\sum_{j=1}^{C} \exp(z_j/T)}$$

$$\Updownarrow$$

$$\sum_{j=1}^{L} \frac{\exp(z_j/\tilde{T})}{\sum_{c=1}^{C} \exp(z_c/\tilde{T})} \geq \sum_{j=1}^{L} \frac{\exp(z_j/T)}{\sum_{c=1}^{C} \exp(z_c/T)}$$

as stated in the theorem. Note that the inequality is strict unless $L = C$ (both sides equal 1) or $T = \tilde{T}$ or $z_1 = \ldots = z_C$ (all the pairs are equal).

$\square$

**Corollary A.2.** *The threshold value $\hat{q}_T$ of APS and RAPS decreases monotonically as the temperature $T$ increases.*

*Proof.* Let us start with APS. Set $T \geq \tilde{T}$. For each sample $(\mathbf{x}, y)$ in the CP set, we get the post softmax vectors $\hat{\boldsymbol{\pi}}_T$ and $\hat{\boldsymbol{\pi}}$, with and without TS, respectively. By Theorem 4.1, applied on the sorted vector $\boldsymbol{\pi} = [\hat{\pi}_{(1)}, \ldots, \hat{\pi}_{(C)}]^{\top}$ with $L = L_y$ (the index that y is permuted to after sorting), we have that

$$\sum_{j=1}^{L_y} \pi_{\tilde{T},j} \geq \sum_{j=1}^{L_y} \pi_{T,j} \tag{8}$$

That is, the score of APS decreases *universally* for each sample in the CP set. This implies that $\hat{q}$, the $\frac{\lceil (n+1)(1-\alpha) \rceil}{n}$ quantile of the scores of the samples of the CP set, decreases as well.

We turn to consider RAPS. In this case, a decrease due to TS in the score of each sample $(\mathbf{x}, y)$ in the CP set, i.e.,

$$\sum_{i=1}^{L_y} \hat{\pi}_{\tilde{T},(i)}(\mathbf{x}) + \lambda(L_y - k_{reg})_+ \geq \sum_{i=1}^{L_y} \hat{\pi}_{T,(i)}(\mathbf{x}) + \lambda(L_y - k_{reg})_+,$$

simply follows from adding $\lambda(L_y - k_{reg})_+$ to both sides of (8). The rest of the arguments are exactly as in APS.

$\square$

**Proposition A.3.** *Let $\mathbf{z} \in \mathbb{R}^C$ such that $z_1 \geq z_2 \geq \cdots \geq z_C$, $\hat{\boldsymbol{\pi}}_T = \boldsymbol{\sigma}(\mathbf{z}/T)$ and $\hat{\boldsymbol{\pi}} = \boldsymbol{\sigma}(\mathbf{z})$. Let $\hat{q}, \hat{q}_T \in (0, 1]$ such that if $T > 1$ then $\hat{q} \geq \hat{q}_T$ and if $0 < T < 1$ then $\hat{q} \leq \hat{q}_T$. Let $L = \min\{l : \sum_{i=1}^{l} \hat{\pi}_i \geq \hat{q}\}$, $L_T = \min\{l : \sum_{i=1}^{l} \hat{\pi}_{T,i} \geq \hat{q}_T\}$. The following holds:*

$$\begin{cases} \text{If } 0 < T < 1 \text{ and } L_T > 1 \quad \text{then}: \quad \sum_{i=1}^{L_T-1} \hat{\pi}_i - \sum_{i=1}^{L_T-1} \hat{\pi}_{T,i} \leq \hat{q} - \hat{q}_T \implies L \geq L_T \\ \text{If } T > 1 \text{ and } L > 1 \quad \text{then}: \quad \sum_{i=1}^{L-1} \hat{\pi}_i - \sum_{i=1}^{L-1} \hat{\pi}_{T,i} \geq \hat{q} - \hat{q}_T \implies L \leq L_T \end{cases} \tag{9}$$

*Proof.* We start with **T > 1 branch**, for which $\hat{q} \geq \hat{q}_T$:

$$\sum_{i=1}^{L-1} \hat{\pi}_i - \sum_{i=1}^{L-1} \hat{\pi}_{T,i} \geq \hat{q} - \hat{q}_T \iff \sum_{i=1}^{L-1} \hat{\pi}_i \geq \sum_{i=1}^{L-1} \hat{\pi}_{T,i} + \hat{q} - \hat{q}_T \qquad (10)$$

Note that $L := \min\{\ell : \sum_{i=1}^{\ell} \hat{\pi}_i \geq \hat{q}\} \implies \sum_{i=1}^{L-1} \hat{\pi}_i < \hat{q}$. Thus, if (10) holds, then:

$$\hat{q} > \sum_{i=1}^{L-1} \hat{\pi}_{T,i} + \hat{q} - \hat{q}_T \iff \hat{q}_T > \sum_{i=1}^{L-1} \hat{\pi}_{T,i}.$$

Therefore, $L_T := \min\{\ell : \sum_{i=1}^{\ell} \hat{\pi}_{T,i} \geq \hat{q}_T\}$ cannot be smaller than $L$.

We continue with **0 < T < 1 branch**, for which $\hat{q}_T \geq \hat{q}$, we will take similar steps:

$$\sum_{i=1}^{L_T-1} \hat{\pi}_i - \sum_{i=1}^{L_T-1} \hat{\pi}_{T,i} \leq \hat{q} - \hat{q}_T \iff \sum_{i=1}^{L_T-1} \hat{\pi}_{T,i} \geq \sum_{i=1}^{L_T-1} \hat{\pi}_i + \hat{q}_T - \hat{q} \qquad (11)$$

Note that $L_T := \min\{\ell : \sum_{i=1}^{\ell} \hat{\pi}_{T,i} \geq \hat{q}_T\} \implies \sum_{i=1}^{L_T-1} \hat{\pi}_{T,i} < \hat{q}_T$. Thus, if (11) holds, then:

$$\hat{q}_T > \sum_{i=1}^{L_T-1} \hat{\pi}_i + \hat{q}_T - \hat{q} \iff \hat{q} > \sum_{i=1}^{L_T-1} \hat{\pi}_i.$$

Therefore, $L := \min\{\ell : \sum_{i=1}^{\ell} \hat{\pi}_i \geq \hat{q}\}$ cannot be smaller than $L_T$.

$\square$

**Theorem A.4.** *Let $\mathbf{z} \in \mathbb{R}^C$ such that $z_1 \geq z_2 \geq \cdots \geq z_C$ and denote $\Delta z = z_1 - z_2$. Then, the following holds:*

$$\begin{cases} If \quad 0 < T < 1: \quad \Delta z > \max\left\{\dfrac{T}{T-1}\ln\left(\dfrac{T}{4}\right), \dfrac{T}{T+1}\ln\left(\dfrac{4(C-1)^2}{T}\right)\right\} \implies \nabla_{z_1} g(\mathbf{z}; T, M) > 0 \\ If \quad T > 1: \quad \Delta z > \max\left\{\dfrac{T}{T-1}\ln(4T), \dfrac{T}{T+1}\ln(4T(C-1)^2)\right\} \implies \nabla_{z_1} g(\mathbf{z}; T, M) < 0 \end{cases} \qquad (12)$$

*Proof.* Let us start with **T > 1 branch:** The gap function is defined as follows:

$$g_z(\mathbf{z}; T, M) = \frac{\sum_{i=1}^{M} \exp(z_i)}{\sum_{j=1}^{C} \exp(z_j)} - \frac{\sum_{i=1}^{M} \exp(z_i/T)}{\sum_{j=1}^{C} \exp(z_j/T)}$$

Let us differentiate with respect to $z_1$

$$\nabla_{z_1} g_z(\mathbf{z}; T, M) = \frac{\exp(z_1)\left[\sum_{j=1}^{C}\exp(z_j) - \sum_{i=1}^{M}\exp(z_i)\right]}{\left[\sum_{j=1}^{C}\exp(z_j)\right]^2} - \frac{\frac{1}{T}\exp(z_1/T)\left[\sum_{j=1}^{C}\exp(z_j/T) - \sum_{i=1}^{M}\exp(z_i/T)\right]}{\left[\sum_{j=1}^{C}\exp(z_j/T)\right]^2}$$

$$= \frac{\exp(z_1)\sum_{c=M+1}^{C}\exp(z_c)}{\left[\sum_{c=1}^{C}\exp(z_c)\right]^2} - \frac{\frac{1}{T}\exp(z_1/T)\sum_{c=M+1}^{C}\exp(z_c/T)}{\left[\sum_{c=1}^{C}\exp(z_c/T)\right]^2}$$

Therefore,

$$\nabla_{z_1} g_z(\mathbf{z}; T, M) < 0 \iff \exp\left(z_1\left(1 - \frac{1}{T}\right)\right) < \frac{1}{T} \frac{\sum_{c=M+1}^{C} \exp(z_c/T)}{\sum_{c=M+1}^{C} \exp(z_c)} \left[\frac{\sum_{c=1}^{C} \exp(z_c)}{\sum_{c=1}^{C} \exp(z_c/T)}\right]^2 \tag{13}$$

where we arranged the inequality and used $\exp(z_1)/\exp(z_1/T) = \exp\left(z_1\left(1 - \frac{1}{T}\right)\right)$.

According to the auxiliary lemma in Theorem A.1, if we substitute $\tilde{T} = 1$ we get: for all $z_i > z_j$ and $T > 1$ we have $\exp(z_i) \cdot \exp(z_j/T) > \exp(z_i/T) \cdot \exp(z_j)$.

Therefore, taking $i = M + 1$ and summing both sides of the inequality over $j = M + 1, ..., C$, the following holds:

$$\frac{\sum_{c=M+1}^{C} \exp(z_c/T)}{\sum_{c=M+1}^{C} \exp(z_c)} > \frac{\exp(z_{M+1}/T)}{\exp(z_{M+1})} \tag{14}$$

In addition, note that $\forall A$ such that $A > \max\left(2(C-1)\exp(-\Delta z/T), 2\right)$ we get:

$$A > (C-1)\exp(-\Delta z/T) + 1 \implies A\exp(z_1/T) > \sum_{c=1}^{C} \exp(z_c/T) \tag{15}$$

where the implication follows from

$$\sum_{c=1}^{C} \exp(z_c/T)/\exp(z_1/T) = 1 + \sum_{c=2}^{C} \exp(-(z_1 - z_c)/T) \le 1 + (C-1)\exp(-(z_1 - z_2)/T).$$

Using the above inequalities ((14) and (15)) we obtain

$$\frac{1}{T} \frac{\sum_{c=M+1}^{C} \exp(z_c/T)}{\sum_{c=M+1}^{C} \exp(z_c)} \left[\frac{\sum_{c=1}^{C} \exp(z_c)}{\sum_{c=1}^{C} \exp(z_c/T)}\right]^2 > \frac{1}{T} \frac{\exp(z_{M+1}/T)}{\exp(z_{M+1})} \left[\frac{\exp(z_1)}{A\exp(z_1/T)}\right]^2 \ge \frac{1}{A^2 T} \exp\left((2z_1 - z_2)\left(1 - \frac{1}{T}\right)\right)$$

where in the last inequality we used

$$\frac{\exp(z_{M+1}/T)}{\exp(z_{M+1})} \frac{\exp(z_1)}{\exp(z_1/T)} = \exp\left((z_1 - z_{M+1})\left(1 - \frac{1}{T}\right)\right) \ge \exp\left((z_1 - z_2)\left(1 - \frac{1}{T}\right)\right).$$

Hence, according to (13): $\exp\left(z_1\left(1 - \frac{1}{T}\right)\right) < \frac{1}{A^2 T} \exp\left((2z_1 - z_2)\left(1 - \frac{1}{T}\right)\right) \implies \nabla_{z_1} g_z(\mathbf{z}; T, M) < 0$

Note that

$$\exp\left(z_1\left(1 - \frac{1}{T}\right)\right) < \frac{1}{A^2 T} \exp\left((2z_1 - z_2)\left(1 - \frac{1}{T}\right)\right)$$

$$\iff 1 < \frac{1}{A^2 T} \exp\left((z_1 - z_2)\left(1 - \frac{1}{T}\right)\right) \iff \Delta z > \frac{T}{T-1} \ln(A^2 T)$$

And by using the definition of $A$ we obtain:

$$\Delta z > \max\left(\frac{T}{T-1} \ln(4T), \frac{T}{T+1} \ln(4T(C-1)^2)\right) \implies \nabla_{z_1} g_z(\mathbf{z}; T, M) < 0$$

We continue with **$0 < T < 1$ branch:** Based on steps we took for the previous branch:

$$\nabla_{z_1} g_z(\mathbf{z}; T, M) > 0 \iff \exp\left(z_1\left(1 - \frac{1}{T}\right)\right) > \frac{1}{T} \frac{\sum_{c=M+1}^{C} \exp(z_c/T)}{\sum_{c=M+1}^{C} \exp(z_c)} \left[\frac{\sum_{c=1}^{C} \exp(z_c)}{\sum_{c=1}^{C} \exp(z_c/T)}\right]^2 \tag{16}$$

Note that according to the auxiliary lemma in Theorem A.1, if we substitute $\tilde{T} = 1$ we get: for all $z_i > z_j$ and $0 < T < 1$ we have $\exp(z_i) \cdot \exp(z_j/T) > \exp(z_i/T) \cdot \exp(z_j)$ and therefore the following holds:

$$\frac{\sum_{c=M+1}^{C} \exp(z_c)}{\sum_{c=M+1}^{C} \exp(z_c/T)} > \frac{\exp(z_{M+1})}{\exp(z_{M+1}/T)}$$

In addition, note that $\forall A$ such that $A > \max\left(2(C-1)\exp(-\Delta z), 2\right)$ we get:

$$A > (C-1)\exp(-\Delta z) + 1 \implies A\exp(z_1) > \sum_{c=1}^{C} \exp(z_c) \tag{17}$$

Using above inequalities we obtain

$$\frac{1}{T}\frac{\sum_{c=M+1}^{C}\exp(z_c/T)}{\sum_{c=M+1}^{C}\exp(z_c)}\left[\frac{\sum_{c=1}^{C}\exp(z_c)}{\sum_{c=1}^{C}\exp(z_c/T)}\right]^2 < \frac{1}{T}\frac{\exp(z_{M+1}/T)}{\exp(z_{M+1})}\left[\frac{A\exp(z_1)}{\exp(z_1/T)}\right]^2 \leq \frac{A^2}{T}\exp\left((2z_1 - z_2)\left(1 - \frac{1}{T}\right)\right)$$

Hence, according to (16): $\exp\left(z_1\left(1 - \frac{1}{T}\right)\right) > \frac{A^2}{T}\exp\left((2z_1 - z_2)\left(1 - \frac{1}{T}\right)\right) \implies \nabla_{z_1}g_z(\mathbf{z}; T, M) > 0$

Note that

$$\exp\left(z_1\left(1 - \frac{1}{T}\right)\right) > \frac{A^2}{T}\exp\left((2z_1 - z_2)\left(1 - \frac{1}{T}\right)\right) \iff \Delta z > \frac{T}{T-1}\ln\left(\frac{T}{A^2}\right)$$

The sign of the in-equality changed because $1 - \frac{1}{T} < 0$.

And by using the definition of $A$ we obtain:

$$\Delta z > \max\left(\frac{T}{T-1}\ln\left(\frac{T}{4}\right), \frac{T}{T+1}\ln\left(\frac{4(C-1)^2}{T}\right)\right) \implies \nabla_{z_1}g_z(\mathbf{z}; T, M) > 0$$

$\square$

**Proposition A.5.** *Let $\mathbf{z} \in \mathbb{R}^C$, $\boldsymbol{\sigma}(\cdot)$ be the softmax function, and $\Delta^{C-1}$ denote the simplex in $\mathbb{R}^C$. Consider Shannon's entropy $H : \Delta^{C-1} \to \mathbb{R}$, i.e., $H(\boldsymbol{\pi}) = -\sum_{i=1}^{C} \pi_i \ln(\pi_i)$. Unless $\mathbf{z} \propto \mathbf{1}_C$ (then $\boldsymbol{\sigma}(\mathbf{z}/T) = \boldsymbol{\sigma}(\mathbf{z})$), we have that $H(\boldsymbol{\sigma}(\mathbf{z}/T))$ is strictly monotonically increasing as $T$ grows.*

*Proof.* To prove this statement, let us show that the function $f(T) = H(\boldsymbol{\sigma}(\mathbf{z}/T))$ monotonically increases (as $T$ increases, regardless of $\mathbf{z}$). To achieve this, we need to show that $f'(T) = \frac{d}{dT}H(\boldsymbol{\sigma}(\mathbf{z}/T)) \geq 0$.

By the chain-rule, $f'(T) = \frac{d}{dT}\left(H(\boldsymbol{\sigma}(\mathbf{z}/T))\right) = \frac{\partial H(\boldsymbol{\sigma})}{\partial \boldsymbol{\sigma}}\frac{\partial \boldsymbol{\sigma}(\mathbf{z})}{\partial \mathbf{z}}\frac{\partial}{\partial T}(\mathbf{z}/T)$. Let us compute each term:

$$\frac{\partial H(\boldsymbol{\sigma})}{\partial \sigma_i} = -\ln(\sigma_i) - \frac{1}{\sigma_i}\cdot\sigma_i = -\ln(\sigma_i) - 1 \implies \frac{\partial H(\boldsymbol{\sigma})}{\partial \boldsymbol{\sigma}} = -\ln(\boldsymbol{\sigma})^\top - \mathbf{1}_C^\top$$

$$\frac{\partial \sigma_i(\mathbf{z})}{\partial z_j} = \sigma_i(\mathbf{z})\cdot(\mathbb{1}\{i = j\} - \sigma_j(\mathbf{z})) \implies \frac{\partial \boldsymbol{\sigma}(\mathbf{z})}{\partial \mathbf{z}} = \text{diag}(\boldsymbol{\sigma}(\mathbf{z})) - \boldsymbol{\sigma}(\mathbf{z})\boldsymbol{\sigma}(\mathbf{z})^\top$$

$$\frac{\partial}{\partial T}(\mathbf{z}/T) = -\frac{1}{T^2}\mathbf{z}$$

where in $\ln(\boldsymbol{\sigma})$ the function operates entry-wise and $\mathbb{1}\{i = j\}$ is the indicator function (equals 1 if $i = j$ and 0 otherwise). Next, observe that $\mathbf{1}_C^\top \left( \text{diag}(\boldsymbol{\sigma}) - \boldsymbol{\sigma}\boldsymbol{\sigma}^\top \right) = \boldsymbol{\sigma}^\top - \boldsymbol{\sigma}^\top = \mathbf{0}^\top$. Consequently, we get

$$
\begin{aligned}
\frac{d}{dT}\left(H(\boldsymbol{\sigma}(\mathbf{z}/T))\right) &= \frac{1}{T^2} \ln(\boldsymbol{\sigma}(\mathbf{z}))^\top (\text{diag}(\boldsymbol{\sigma}(\mathbf{z})) - \boldsymbol{\sigma}(\mathbf{z})\boldsymbol{\sigma}(\mathbf{z})^\top)\mathbf{z} \\
&= \frac{1}{T^2}(\mathbf{z} - s(\mathbf{z})\mathbf{1}_C)^\top (\text{diag}(\boldsymbol{\sigma}(\mathbf{z})) - \boldsymbol{\sigma}(\mathbf{z})\boldsymbol{\sigma}(\mathbf{z})^\top)\mathbf{z} \\
&= \frac{1}{T^2}\mathbf{z}^\top (\text{diag}(\boldsymbol{\sigma}(\mathbf{z})) - \boldsymbol{\sigma}(\mathbf{z})\boldsymbol{\sigma}(\mathbf{z})^\top)\mathbf{z}
\end{aligned}
$$

where in the second equality we used $[\ln(\boldsymbol{\sigma}(\mathbf{z}))]_i = \ln\left( \frac{\exp(z_i)}{\sum_{j=1}^C \exp(z_j)} \right) = z_i - s(\mathbf{z})$.

Therefore, for establishing that $\frac{d}{dT}(H(\boldsymbol{\sigma}(\mathbf{z}/T))) \geq 0$, we can show that $(\text{diag}(\boldsymbol{\sigma}) - \boldsymbol{\sigma}\boldsymbol{\sigma}^\top)$ is a positive semi-definite matrix. Let $\tilde{\sigma}_i = \exp(z_i)$ and notice that $\sigma_i = \frac{\tilde{\sigma}_i}{\sum_{j=1}^C \tilde{\sigma}_j}$. Indeed, for any $\mathbf{u} \in \mathbb{R}^C \setminus \{\mathbf{0}\}$ we have that

$$
\begin{aligned}
\mathbf{u}^\top (\text{diag}(\boldsymbol{\sigma}) - \boldsymbol{\sigma}\boldsymbol{\sigma}^\top)\mathbf{u} &= \sum_{i=1}^C u_i^2 \sigma_i - \left( \sum_{i=1}^C u_i \sigma_i \right)^2 \\
&= \frac{\sum_{i=1}^C u_i^2 \tilde{\sigma}_i \cdot \sum_{j=1}^C \tilde{\sigma}_j - \left( \sum_{i=1}^C u_i \tilde{\sigma}_i \right)^2}{\left( \sum_{j=1}^C \tilde{\sigma}_j \right)^2} \\
&\geq 0,
\end{aligned}
$$

where the inequality follows from Cauchy–Schwarz inequality: $\displaystyle\sum_{i=1}^C u_i \tilde{\sigma}_i = \sum_{i=1}^C u_i \sqrt{\tilde{\sigma}_i}\sqrt{\tilde{\sigma}_i} \leq \sqrt{\sum_{i=1}^C u_i^2 \tilde{\sigma}_i}\sqrt{\sum_{j=1}^C \tilde{\sigma}_j}$.

Cauchy–Schwarz inequality is attained with equality iff $u_i \sqrt{\tilde{\sigma}_i} = c\sqrt{\tilde{\sigma}_i}$ with the same constant $c$ for $i = 1, \ldots, C$, i.e., when $\mathbf{u} = c\mathbf{1}_C$. Recalling that $\frac{d}{dT}(H(\boldsymbol{\sigma}(\mathbf{z}/T))) = \frac{1}{T^2}\mathbf{z}^\top(\text{diag}(\boldsymbol{\sigma}) - \boldsymbol{\sigma}\boldsymbol{\sigma}^\top)\mathbf{z}$, this implies that $\frac{d}{dT}(H(\boldsymbol{\sigma}(\mathbf{z}/T))) = 0 \iff z_1 = \ldots = z_C$, and otherwise $\frac{d}{dT}(H(\boldsymbol{\sigma}(\mathbf{z}/T))) > 0$.

$\square$

## A.1. On the link between (3) and (5)

With our assumption that $\hat{q}$ and $\hat{q}_T$ are associated with the same quantile sample $\mathbf{z}^q$, we have that $\hat{q} - \hat{q}_T$ in (3) can be written as $g(\mathbf{z}^q; T, L^q)$, where $L^q$ denotes the rank of the true label of $\mathbf{z}^q$. The left-hand side in (3) can be written as $g(\mathbf{z}; T, L_T - 1)$ for $T < 1$ and $g(\mathbf{z}; T, L - 1)$ for $T > 1$. In this section, we theoretically show the similarity between $g(\mathbf{z}; T, M)$ and $g(\mathbf{z}; T, L_q) \ \forall M \in [C]$. Specifically, we prove exponential decay of this distance as function of $\Delta z$. According to Figure 2, this justifies the assumption of $g(\mathbf{z}^q, T, M) \approx g(\mathbf{z}^q, T, L^q)$, which leads to studying (5).

**Proposition A.6.** *Let $\mathbf{z} \in \mathbb{R}^C$ such that $z_1 \geq z_2 \geq \cdots \geq z_C$. Consider the following functions:*

$$
d(\mathbf{z}; T, i) = \frac{\exp(z_i)}{\sum_{j=1}^C \exp(z_j)} - \frac{\exp(z_i/T)}{\sum_{j=1}^C \exp(z_j/T)}
$$

$$
g(\mathbf{z}; T, M) = \frac{\sum_{i=1}^M \exp(z_i)}{\sum_{j=1}^C \exp(z_j)} - \frac{\sum_{i=1}^M \exp(z_i/T)}{\sum_{j=1}^C \exp(z_j/T)} = \sum_{i=1}^M d(\mathbf{z}; T, i)
$$

*Let $s_{T>1} = \max\{i \in [C] : d(\mathbf{z}; T, j) > 0\}$, i.e., the last index where $d(\mathbf{z}, T, \cdot)$ is positive. Similarly, let $s_{T<1} =$*

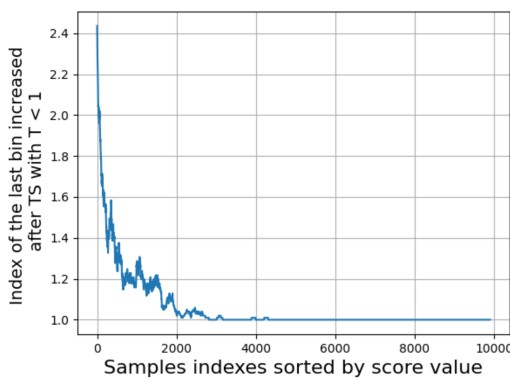 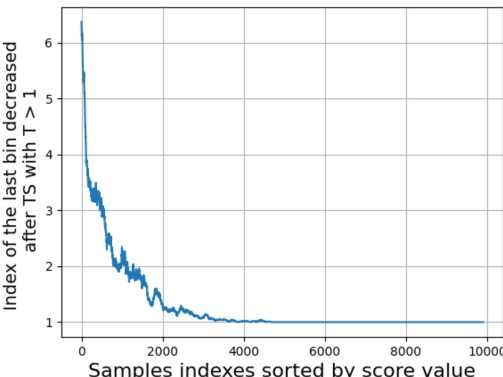

*Figure 5.* Considering CIFAR100-ResNet50 with sorted logits vectors. Left: the last index increased by TS with $T = 0.5$; Right: the last index decreased by TS with $T = 1.5$, for each sample sorted by score value. We see that $s_{T<1} = 1$ and $s_{T>1} = 1$ for the quantile sample.

$\max \{i \in [C] : d(\mathbf{z}; T, j) < 0\}$, *i.e., the last index where $d(\mathbf{z}, T, \cdot)$ is negative. The following holds:*

$$\begin{cases} \text{If} \quad 0 < T < 1: \quad s_{T<1} = 1 \implies \forall M, L^q \in [C] \quad |g(\mathbf{z}; T, M) - g(\mathbf{z}; T, L^q)| < \dfrac{(C-1)\exp(-\Delta z)}{(C-1)\exp(-\Delta z) + 1} \\[3mm] \text{If} \quad T > 1: \quad s_{T>1} = 1 \implies \forall M, L^q \in [C] \quad |g(\mathbf{z}; T, M) - g(\mathbf{z}; T, L^q)| < \dfrac{(C-1)\exp(-\Delta z/T)}{(C-1)\exp(-\Delta z/T) + 1} \end{cases}$$

When applying the proposition on the quantile sample $\mathbf{z}^q$, since it has $\Delta z^q \gg 1$ we get small upper bounds.

As can be seen in the proposition, the bounds require that $s_{T<1}$ and $s_{T=1}$ equal 1. Recall also that the logits vector is sorted. For $T > 1$ (resp. $T < 1$), this means that the TS attenuates (resp. amplifies) only the maximal softmax bin. This is expected to hold for the quantile sample, due to having a very dominant entry in $\mathbf{z}^q$. We show it empirically in Figure 5, which presents the curves for $s_{T<1}$ and $s_{T>1}$ across samples sorted by their scores for CIFAR100-ResNet50. Both curves indicate that approximately the first third of the samples correspond to $s > 1$, while higher-score samples align with $s = 1$. Notably, for the quantile sample $\mathbf{z}^q$ —characterized by scores exceeding 90% of the samples — we observe that $s_{T<1} = 1$ and $s_{T>1} = 1$.

**Proof of Proposition A.6**

We start by stating and proving Lemmas A.7 and A.8, which serve as auxiliary to Proposition A.6.

**Lemma A.7.** *Let $\boldsymbol{\pi}$ be a descendingly sorted softmax vector and $\boldsymbol{\pi}_T$ the same vector after temperature scaling. Define the difference vector $\mathbf{d} := \boldsymbol{\pi} - \boldsymbol{\pi}_T$. Then, there exists an index such that the vector $\mathbf{d}$ is partitioned into two segments, where all elements in one segment have the opposite sign to those in the other segment.*

*Proof.* First, let us formulate this proposition:
Let $\mathbf{z} \in \mathbb{R}^C$ such that $z_1 \geq z_2 \geq \cdots \geq z_C$. Consider the following function difference:

$$d(\mathbf{z}; T, i) = \frac{\exp(z_i)}{\sum_{j=1}^{C} \exp(z_j)} - \frac{\exp(z_i/T)}{\sum_{j=1}^{C} \exp(z_j/T)} = \pi_i - \pi_{T,i}$$

The following holds:

$$\begin{cases} \text{If} \quad 0 < T < 1 \quad \text{and} \quad \exists i \in [C] \quad \text{s.t.} \quad d(\mathbf{z}; T, i) > 0: \quad \forall k > i \quad d(\mathbf{z}; T, k) > 0 \\ \text{If} \quad T > 1 \quad \text{and} \quad \exists i \in [C] \quad \text{s.t.} \quad d(\mathbf{z}; T, i) < 0: \quad \forall k > i \quad d(\mathbf{z}; T, k) < 0 \end{cases}$$

Denote $A := \dfrac{1}{\sum_{j=1}^{C} \exp(z_j)}$ and $B := \dfrac{1}{\sum_{j=1}^{C} \exp(z_j/T)}$. Notice that $\mathbf{z}$ and $T$ are constants and only $i$ is changing in the

theorem condition. Let us rewrite $d(\mathbf{z}; T, i)$:

$$d(\mathbf{z}; T, i) = d(z_i) = A \exp(z_i) - B \exp(z_i/T)$$

Notice that $d(z_i)$ is a discrete function because $z_i \in \mathbf{z}$. To understand when is it possible that $d(z_i) = 0$ we convert the function to continuous, i.e., $z_i$ can be any value. The continuous function (that gets single variable instead of vector) is as follows:

$$d(x) = A \exp(x) - B \exp(x/T)$$

There is a *single* solution of the equation $d(x) = 0$ which is $x = \dfrac{T}{T-1} \ln(B/A) := x^*$. We will now divide the rest of the proof into two temperature branches:

- **T > 1 branch:** Assume $x_1 < x^*, d(x_1) < 0$ then, $\forall x < x_1$ we have $d(x) < 0$.
  In our discrete case, we know that $d(z_1) > 0$ (substitute $L = 1$ in Theorem A.1), and that $d(z_i) < 0$ (therefore $z_i < z^*$ in our analogy to the continuous case), consequently $\forall z_k < z_i,\ d(z_k) < 0$. Overall, because $k > i$, $z_k < z_i$ and we get that $d(z; T, k) < 0$.

- **0 < T < 1 branch:** Assume $x_2 > x^*, d(x_2) > 0$ then, $\forall x > x_2$ we have $d(x) > 0$.
  In our discrete case, we know that $d(z_1) < 0$ (substitute $L = 1$ in Theorem A.1), and that $d(z_i) > 0$ (therefore $z_i < z^*$ in our analogy to the continuous case), consequently $\forall z_k < z_i,\ d(z_k) > 0$. Overall, because $k > i$, $z_k < z_i$ and we get that $d(z; T, k) > 0$.

$\square$

**Lemma A.8.** *Let $\mathbf{z} \in \mathbb{R}^C$ such that $z_1 \geq z_2 \geq \cdots \geq z_C$. Consider the following functions:*

$$d(\mathbf{z}; T, i) = \frac{\exp(z_i)}{\sum_{j=1}^{C} \exp(z_j)} - \frac{\exp(z_i/T)}{\sum_{j=1}^{C} \exp(z_j/T)}$$

$$g(\mathbf{z}; T, M) = \frac{\sum_{i=1}^{M} \exp(z_i)}{\sum_{j=1}^{C} \exp(z_j)} - \frac{\sum_{i=1}^{M} \exp(z_i/T)}{\sum_{j=1}^{C} \exp(z_j/T)} = \sum_{i=1}^{M} d(\mathbf{z}; T, i)$$

*Then, if we denote by $s_{T>1} = \max\{i \in [C] : d(\mathbf{z}; T, j) > 0\}$ the last index where $d(\mathbf{z}, T, \cdot)$ is positive and similarly by $s_{T<1} = \max\{i \in [C] : d(\mathbf{z}; T, j) < 0\}$ the last index where $d(\mathbf{z}, T, \cdot)$ is negative, the following holds:*

$$
\begin{cases}
\text{If } \ 0 < T < 1: \quad \forall M, L^q \in [C] \quad |g(\mathbf{z}; T, M) - g(z; T, L^q)| < \left| \sum_{i=1}^{s_{T<1}} d(z^q; T, i) \right| \\[4mm]
\text{If } \ T > 1: \quad \forall M, L^q \in [C] \quad |g(\mathbf{z}; T, M) - g(\mathbf{z}; T, L^q)| < \left| \sum_{i=1}^{s_{T>1}} d(z^q; T, i) \right|
\end{cases}
$$

*Proof.* Let us begin by rewriting $|g(\mathbf{z}; T, M) - g(\mathbf{z}; T, L^q)|$, assume $M > L^q$ without loss of generality:

$$|g(\mathbf{z}; T, M) - g(\mathbf{z}; T, L^q)| = \left| \frac{\sum_{i=1}^{M} \exp(z_i)}{\sum_{j=1}^{C} \exp(z_j)} - \frac{\sum_{i=1}^{M} \exp(z_i/T)}{\sum_{j=1}^{C} \exp(z_j/T)} - \left( \frac{\sum_{i=1}^{L^q} \exp(z_i)}{\sum_{j=1}^{C} \exp(z_j)} - \frac{\sum_{i=1}^{L^q} \exp(z_i/T)}{\sum_{j=1}^{C} \exp(z_j/T)} \right) \right|$$

$$= \left| \sum_{i=1}^{M} d(\mathbf{z}; T, i) - \sum_{i=1}^{L^q} d(\mathbf{z}; T, i) \right| = \left| \sum_{i=L^q+1}^{M} d(\mathbf{z}; T, i) \right|$$

Henceforth we denote $s := s_{T>1}$ or $s := s_{T<1}$ depending on the temperature. By Lemma A.7, $\forall i < s$ we have $d(\mathbf{z}; T, i) > 0$ and $\forall i > s$ we have $d(\mathbf{z}; T, i) < 0$, therefore we can divide the analysis of $\left| \sum_{i=L^q+1}^{M} d(\mathbf{z}; T, i) \right|$ into 3 cases:

- $L^q + 1 \geq s$ - in this case, we sum only on differences with the same sign, and we can create an upper bound for this case by summing all the differences with the same sign:

$$\left| \sum_{i=L^q+1}^{M} d(\mathbf{z}; T, i) \right| \leq \left| \sum_{i=s+1}^{C} d(\mathbf{z}; T, i) \right|$$

- $M \leq s$ - in this case like previous case, we sum only on differences with the same sign, and we can create an upper bound for this case by summing all the differences with the same sign, note that in any case the starting index of the summation is 2 or higher ($L^q + 1 \geq 2$), therefore we can exclude the first difference:

$$\left| \sum_{i=L^q+1}^{M} d(\mathbf{z}; T, i) \right| \leq \left| \sum_{i=2}^{s} d(\mathbf{z}; T, i) \right|$$

- $L^q + 1 < s$ and $M > s$ - in this case, we sum both on positive and negative differences, and we can create an upper bound for this case by taking the maximum between previous cases bounds:

$$\left| \sum_{i=L^q+1}^{M} d(\mathbf{z}; T, i) \right| \leq \max \left\{ \left| \sum_{i=s+1}^{C} d(\mathbf{z}; T, i) \right|, \left| \sum_{i=2}^{s} d(\mathbf{z}; T, i) \right| \right\}$$

Overall , we can write the upper bound of $\left| \sum_{i=L^q+1}^{M} d(\mathbf{z}; T, i) \right|$ as in case 3:

$$\left| \sum_{i=L^q+1}^{M} d(\mathbf{z}; T, i) \right| \leq \max \left\{ \left| \sum_{i=s+1}^{C} d(\mathbf{z}; T, i) \right|, \left| \sum_{i=2}^{s} d(\mathbf{z}; T, i) \right| \right\}$$

Note that the summation of all differences is zero, I.e. $\sum_{i=1}^{C} d(\mathbf{z}; T, i) = 0$, therefore,

$\sum_{i=1}^{s} d(\mathbf{z}; T, i) = - \sum_{i=s+1}^{C} d(\mathbf{z}; T, i)$ and $\left| \sum_{i=1}^{s} d(\mathbf{z}; T, i) \right| = \left| \sum_{i=s+1}^{C} d(\mathbf{z}; T, i) \right|$. because both summations contain differences with the same sign, subtracting elements from them decrease them, therefore,

$$\left| \sum_{i=2}^{s} d(\mathbf{z}; T, i) \right| < \left| \sum_{i=s+1}^{C} d(\mathbf{z}; T, i) \right|$$

$$\Downarrow$$

$$\max \left\{ \left| \sum_{i=s+1}^{C} d(\mathbf{z}; T, i) \right|, \left| \sum_{i=2}^{s} d(\mathbf{z}; T, i) \right| \right\} = \left| \sum_{i=s+1}^{C} d(\mathbf{z}; T, i) \right| = \left| \sum_{i=1}^{s} d(\mathbf{z}; T, i) \right|$$

And overall we get:

$$\forall M, L^q \in [C] \quad |d(\mathbf{z}; T, M) - g(\mathbf{z}; T, L^q)| < \left| \sum_{i=1}^{s} d(\mathbf{z}; T, i) \right|$$

$$\square$$

**We now turn to proving Proposition A.6.**

*Proof.* **T > 1 branch:**
Substituting $s = 1$ in Lemma A.8 we get:

$$\forall M, L^q \in [C] \quad |g(\mathbf{z}; T, M) - g(\mathbf{z}; T, L^q)| < \left| \sum_{i=1}^{s=1} d(\mathbf{z}; T, i) \right| = |d(\mathbf{z}; T, 1)| = d(\mathbf{z}; T, 1)$$

We can remove the absolute value because $d(\mathbf{z}; T, 1) > 0$ for $T > 1$.
We continue by bounding $d(\mathbf{z}; T, 1)$:

$$d(\mathbf{z}; T, 1) = \frac{\exp(z_1)}{\sum_{j=1}^{C} \exp(z_j)} - \frac{\exp(z_1/T)}{\sum_{j=1}^{C} \exp(z_j/T)} < 1 - \frac{\exp(z_1/T)}{\sum_{j=1}^{C} \exp(z_j/T)}$$

$$= 1 - \frac{\exp(z_1/T)}{\exp(z_1/T) \left[ \sum_{j=2}^{C} \exp(-(z_1 - z_j)/T) + 1 \right]} = 1 - \frac{1}{\sum_{j=2}^{C} \exp(-(z_1 - z_j)/T) + 1}$$

$$\leq 1 - \frac{1}{(C-1) \exp(-\Delta z/T) + 1} = \frac{(C-1) \exp(-\Delta z/T)}{(C-1) \exp(-\Delta)/T) + 1}$$

Overall, we get:

$$\forall M, L^q \in [C] \quad |g(\mathbf{z}; T, M) - g(\mathbf{z}; T, L^q)| < \frac{(C-1) \exp(-\Delta z/T)}{(C-1) \exp(-\Delta z/T) + 1}$$

**$0 < T < 1$ branch:**
Substituting $s = 1$ in Lemma A.8 we get:

$$\forall M, L^q \in [C] \quad |g(\mathbf{z}; T, M) - g(\mathbf{z}; T, L^q)| < \left| \sum_{i=1}^{s=1} d(\mathbf{z}; T, M) \right| = |d(\mathbf{z}; T, 1)| = -d(\mathbf{z}; T, 1)$$

We can remove the absolute value and add minus sign because $d(z; T, 1) < 0$ for $T > 1$.
We continue by bounding $-d(\mathbf{z}; T, 1)$:

$$|d(\mathbf{z}; T, 1)| = -d(\mathbf{z}; T, 1) = \frac{\exp(z_1/T)}{\sum_{j=1}^{C} \exp(z_j/T)} - \frac{\exp(z_1)}{\sum_{j=1}^{C} \exp(z_j)} < 1 - \frac{\exp(z_1)}{\sum_{j=1}^{C} \exp(z_j)}$$

$$= 1 - \frac{\exp(z_1)}{\exp(z_1) \left[ \sum_{j=2}^{C} \exp(-(z_1 - z_j)) + 1 \right]} = 1 - \frac{1}{\sum_{j=2}^{C} \exp(-(z_1 - z_j)) + 1}$$

$$\leq 1 - \frac{1}{(C-1) \exp(-\Delta z) + 1} = \frac{(C-1) \exp(-\Delta z)}{(C-1) \exp(-\Delta z)) + 1}$$

Overall, we get:

$$\forall M, L^q \in [C] \quad |g(\mathbf{z}; T, M) - g(\mathbf{z}; T, L^q)| < \frac{(C-1) \exp(-\Delta z)}{(C-1) \exp(-\Delta z) + 1}$$

$\square$

# B. Additional Experimental Details and Results

## B.1. Training details

For ImageNet models, we utilized pretrained models from the `TORCHVISION.MODELS` sub-package. For full training details, please refer to the following link:

[https://github.com/pytorch/vision/tree/8317295c1d272e0ba7b2ce31e3fd2c048235fc73/references/classification](https://github.com/pytorch/vision/tree/8317295c1d272e0ba7b2ce31e3fd2c048235fc73/references/classification)

For CIFAR-100 and CIFAR-10 models, we use: Batch size: 128; Epochs: 300; Cross-Entropy loss; Optimizer: SGD; Learning rate: 0.1; Momentum: 0.9; Weight decay: 0.0005.

## B.2. Experiments compute resources

We conducted our experiments using an NVIDIA GeForce GTX 1080 Ti. Given the trained models, each experiment runtime is within a range of minutes.

## B.3. Temperature scaling calibration

As mentioned in Section 2.1, two popular calibration objectives are the Negative Log-Likelihood (NLL) (Hastie et al., 2005) and the Expected Calibration Error (ECE) (Naeini et al., 2015).

NLL, given by $\mathcal{L} = -\sum_{i=1}^{n} \ln(\tilde{\pi}_{y_i}(\mathbf{x}_i))$, measures the cross-entropy between the true conditional distribution of data (one-hot vector associated with $y_i$) and $\tilde{\boldsymbol{\pi}}(\mathbf{x}_i)$.

ECE aims to approximate $\mathbb{E}\left[\left|\mathbb{P}\left(\hat{y}(X) = Y | \tilde{\pi}_{\hat{y}(X)}(X) = p\right) - p\right|\right]$. Specifically, the confidence range $[0, 1]$ is divided into $L$ equally sized bins $\{B_l\}$. Each sample $(\mathbf{x}_i, y_i)$ is assigned to a bin $B_l$ according to $\tilde{\pi}_{y_i}(\mathbf{x}_i)$. The objective is given by $\text{ECE} = \sum_{l=1}^{L} \frac{|B_l|}{n} |\text{acc}(B_l) - \text{conf}(B_l)|$, where $\text{acc}(B_l) = \frac{1}{|B_l|} \sum_{i \in B_l} \mathbb{1}\{\hat{y}(\mathbf{x}_i) = y_i\}$ and $\text{conf}(B_l) = \frac{1}{|B_l|} \sum_{i \in B_l} \hat{\pi}_{y_i}(\mathbf{x}_i)$. Here, $\mathbb{1}(\cdot)$ denotes the indicator function.

### B.3.1. ECE VS NLL MINIMIZATION

Above, we defined the two common minimization objectives for the TS calibration procedure. Throughout the paper, we employed the ECE objective. Here, we justify this choice by demonstrating the proximity of the optimal calibration temperature $T^*$ for both objectives.

Table 3. Optimal Temperature for NLL and ECE objectives

| Dataset-Model | $T^*$ - NLL loss | $T^*$ - ECE loss |
|---|---|---|
| CIFAR-100, ResNet50 | 1.438 | 1.524 |
| CIFAR-100, DenseNet121 | 1.380 | 1.469 |
| ImageNet, ResNet152 | 1.207 | 1.227 |
| ImageNet, DenseNet121 | 1.054 | 1.024 |
| ImageNet, ViT-B/16 | 1.18 | 1.21 |
| CIFAR-10, ResNet50 | 1.683 | 1.761 |
| CIFAR-10, ResNet34 | 1.715 | 1.802 |

Using both objectives, we obtain similar optimal calibration temperatures $T^*$, resulting in minor changes to the values in Tables 1 and 2, presented in Section 3.2. Furthermore, in Section 3.3, we examine the effect of TS over a range of temperatures, which naturally includes both optimal temperatures.

### B.3.2. RELIABILITY DIAGRAMS

The *reliability diagram* (DeGroot & Fienberg, 1983; Niculescu-Mizil & Caruana, 2005) is a graphical depiction of a model before and after calibration. The confidence range $[0, 1]$ is divided into bins and the validation samples (not used in the calibration) are assigned to the bins according to $\hat{\pi}_{\hat{y}(\mathbf{x})}(\mathbf{x})$. The average accuracy (Top-1) is computed per bin. In the case of perfect calibration, the diagram should be aligned with the identity function. Any significant deviation from slope of 1 indicates miscalibration.

Below, we present reliability diagrams for dataset-model pairs examined in our study. We divided the confidence range into 10 bins and displayed the accuracy for each bin as a histogram. The red bars represent the calibration error for each bin.

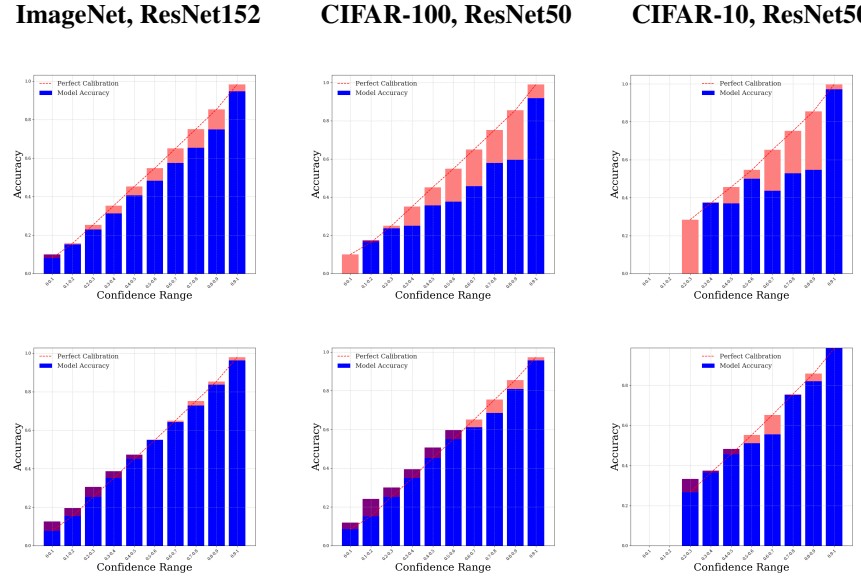

*Figure 6.* Reliability diagrams before (top) and after (bottom) TS calibration with ECE objective.

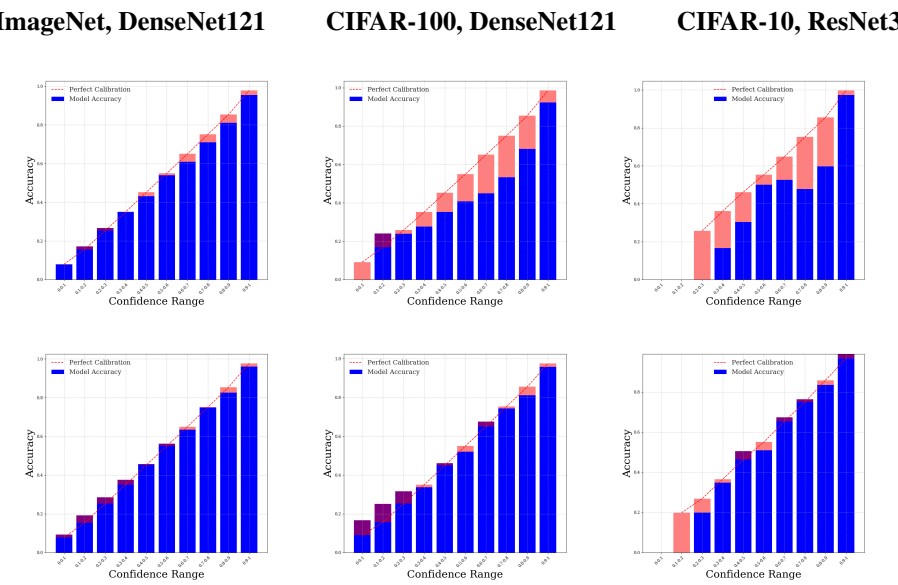

*Figure 7.* Reliability diagrams before (top) and after (bottom) TS calibration with ECE objective.

## B.4. The effect of TS calibration on CP methods

As an extension of Section 3.2, we provide additional experiments where we examine the effect of TS calibration on CP methods with different settings of hyper-parameters ($\alpha$ and CP set size). The tables below present prediction set sizes and coverage metrics before and after TS calibration for different CP set sizes and an additional CP coverage probability value.

*Table 4.* **Prediction Set Size.** AvgSize metric along with $T^*$ and accuracy for dataset-model pairs using LAC, APS, and RAPS algorithms with $\alpha = 0.1$, CP set size 5%, pre- and post-TS calibration.

| Dataset-Model | $T^*$ | Accuracy(%) | | AvgSize | | | AvgSize after TS | | |
| --- | --- | --- | --- | --- | --- | --- | --- | --- | --- |
| | | Top-1 | Top-5 | LAC | APS | RAPS | LAC | APS | RAPS |
| ImageNet, ResNet152 | 1.227 | 78.3 | 94.0 | 1.94 | 7.24 | 3.20 | 1.95 | 10.3 | 4.35 |
| ImageNet, DenseNet121 | 1.024 | 74.4 | 91.9 | 2.70 | 10.1 | 4.71 | 2.77 | 11.2 | 4.91 |
| ImageNet, ViT-B/16 | 1.180 | 83.9 | 97.0 | 2.75 | 10.18 | 2.01 | 2.33 | 19.19 | 2.41 |
| CIFAR-100, ResNet50 | 1.524 | 80.9 | 95.4 | 1.62 | 5.75 | 2.78 | 1.57 | 9.76 | 4.93 |
| CIFAR-100, DenseNet121 | 1.469 | 76.1 | 93.5 | 2.10 | 4.30 | 2.99 | 2.08 | 6.61 | 4.38 |
| CIFAR-10, ResNet50 | 1.761 | 94.6 | 99.7 | 0.92 | 1.05 | 0.95 | 0.91 | 1.13 | 1.01 |
| CIFAR-10, ResNet34 | 1.802 | 95.3 | 99.8 | 0.91 | 1.03 | 0.94 | 0.93 | 1.11 | 1.05 |

*Table 5.* **Prediction Set Size.** AvgSize metric along with $T^*$ and accuracy for dataset-model pairs using LAC, APS, and RAPS algorithms with $\alpha = 0.1$, CP set size 20%, pre- and post-TS calibration.

| Dataset-Model | $T^*$ | Accuracy(%) | | AvgSize | | | AvgSize after TS | | |
| --- | --- | --- | --- | --- | --- | --- | --- | --- | --- |
| | | Top-1 | Top-5 | LAC | APS | RAPS | LAC | APS | RAPS |
| ImageNet, ResNet152 | 1.227 | 78.3 | 94.0 | 1.95 | 7.34 | 3.30 | 1.92 | 12.5 | 4.40 |
| ImageNet, DenseNet121 | 1.024 | 74.4 | 91.9 | 2.73 | 13.1 | 4.70 | 2.76 | 13.3 | 4.88 |
| ImageNet, ViT-B/16 | 1.180 | 83.9 | 97.0 | 2.69 | 10.03 | 1.89 | 2.24 | 19.05 | 2.48 |
| CIFAR-100, ResNet50 | 1.524 | 80.9 | 95.4 | 1.62 | 5.35 | 2.68 | 1.57 | 9.34 | 4.96 |
| CIFAR-100, DenseNet121 | 1.469 | 76.1 | 93.5 | 2.13 | 4.36 | 2.95 | 2.06 | 6.81 | 4.37 |
| CIFAR-10, ResNet50 | 1.761 | 94.6 | 99.7 | 0.91 | 1.04 | 0.98 | 0.91 | 1.13 | 1.05 |
| CIFAR-10, ResNet34 | 1.802 | 95.3 | 99.8 | 0.91 | 1.03 | 0.94 | 0.93 | 1.11 | 1.05 |

*Table 6.* **Coverage Metrics.** MarCovGap and TopCovGap metrics for dataset-model pairs using LAC, APS, and RAPS algorithms with $\alpha = 0.1$, CP set size 5%, pre- and post-TS calibration.

| Dataset-Model | MarCovGap(%) | | | MarCovGap TS(%) | | | TopCovGap(%) | | | TopCovGap TS(%) | | |
| --- | --- | --- | --- | --- | --- | --- | --- | --- | --- | --- | --- | --- |
| | LAC | APS | RAPS | LAC | APS | RAPS | LAC | APS | RAPS | LAC | APS | RAPS |
| ImageNet, ResNet152 | 0 | 0 | 0 | 0 | 0.1 | 0 | 23.5 | 15.7 | 16.9 | 24.1 | 13.6 | 15.0 |
| ImageNet, DenseNet121 | 0 | 0.1 | 0 | 0 | 0 | 0.1 | 24.9 | 15.7 | 18 | 25.2 | 14.9 | 17.6 |
| ImageNet, ViT-B/16 | 0 | 0 | 0.1 | 0.1 | 0 | 0 | 24.1 | 14.9 | 14.5 | 24.8 | 12.4 | 12.6 |
| CIFAR-100, ResNet50 | 0.1 | 0 | 0.1 | 0 | 0.1 | 0 | 13.9 | 11.9 | 10.7 | 12.5 | 8.2 | 7.5 |
| CIFAR-100, DenseNet121 | 0 | 0 | 0.1 | 0 | 0 | 0.1 | 11.6 | 9.5 | 9.0 | 11.7 | 7.8 | 7.7 |
| CIFAR-10, ResNet50 | 0 | 0 | 0 | 0 | 0.1 | 0 | 11.1 | 5.0 | 4.8 | 11.2 | 2.4 | 2.6 |
| CIFAR-10, ResNet34 | 0 | 0.1 | 0.1 | 0 | 0 | 0 | 9.5 | 3.0 | 2.8 | 9.1 | 2.2 | 2.2 |

*Table 7.* **Coverage Metrics.** MarCovGap and TopCovGap metrics for dataset-model pairs using LAC, APS, and RAPS algorithms with $\alpha = 0.1$, CP set size 20%, pre- and post-TS calibration.

| Dataset-Model | MarCovGap(%) | | | MarCovGap TS(%) | | | TopCovGap(%) | | | TopCovGap TS(%) | | |
| --- | --- | --- | --- | --- | --- | --- | --- | --- | --- | --- | --- | --- |
| | LAC | APS | RAPS | LAC | APS | RAPS | LAC | APS | RAPS | LAC | APS | RAPS |
| ImageNet, ResNet152 | 0.1 | 0.1 | 0 | 0.1 | 0 | 0 | 23.6 | 16.3 | 17.5 | 23.6 | 13.9 | 15.6 |
| ImageNet, DenseNet121 | 0 | 0.1 | 0 | 0 | 0 | 0 | 24.9 | 15.7 | 18 | 25.2 | 14.9 | 17.6 |
| ImageNet, ViT-B/16 | 0 | 0 | 0.1 | 0.1 | 0 | 0 | 23.9 | 15.2 | 14.1 | 24.3 | 12.7 | 12.6 |
| CIFAR-100, ResNet50 | 0.1 | 0.1 | 0 | 0 | 0.1 | 0 | 11.4 | 9.9 | 10.0 | 13.0 | 7.7 | 8.2 |
| CIFAR-100, DenseNet121 | 0 | 0 | 0 | 0 | 0 | 0.1 | 11.5 | 9.5 | 9.7 | 12.2 | 7.8 | 8.0 |
| CIFAR-10, ResNet50 | 0 | 0 | 0 | 0 | 0.1 | 0 | 10.8 | 5.1 | 4.6 | 11.0 | 2.1 | 2.6 |
| CIFAR-10, ResNet34 | 0 | 0 | 0.1 | 0 | 0 | 0 | 9.1 | 3.1 | 2.6 | 9.3 | 2.1 | 2.3 |

We can see from the above tables that the results for different CP sizes are very similar. The goal of the CP set is to be large enough to represent the rest of the data from the same distribution. In our experiments, we see that even 5% of the validation set is sufficient for this purpose.

The increased coverage probability is reflected in both prediction set sizes and the conditional coverage metric. We observe an increase in prediction set sizes compared to Table 1, which is expected due to the stricter coverage probability requirement. Note that the tendency for prediction set sizes to increase with $T$ remains. Regarding the coverage metric TopCovGap, we

*Table 8.* **Prediction Set Size.** AvgSize metric along with $T^*$ and accuracy for dataset-model pairs using LAC, APS, and RAPS algorithms with $\alpha = 0.05$, CP set size 10%, pre- and post-TS calibration.

| Dataset-Model | $T^*$ | Accuracy(%) | | AvgSize | | | AvgSize after TS | | |
|---|---|---|---|---|---|---|---|---|---|
| | | Top-1 | Top-5 | LAC | APS | RAPS | LAC | APS | RAPS |
| ImageNet, ResNet152 | 1.227 | 78.3 | 94.0 | 3.28 | 14.9 | 4.10 | 3.22 | 24.1 | 5.1 |
| ImageNet, DenseNet121 | 1.024 | 74.4 | 91.9 | 3.33 | 20.1 | 5.02 | 3.61 | 22.8 | 5.88 |
| ImageNet, ViT-B/16 | 1.180 | 83.9 | 97.0 | 2.91 | 22.80 | 4.51 | 3.02 | 39.8 | 5.55 |
| CIFAR-100, ResNet50 | 1.524 | 80.9 | 95.4 | 3.97 | 11.10 | 3.98 | 2.21 | 16.2 | 6.80 |
| CIFAR-100, DenseNet121 | 1.469 | 76.1 | 93.5 | 4.89 | 8.81 | 5.01 | 4.23 | 12.16 | 6.11 |
| CIFAR-10, ResNet50 | 1.761 | 94.6 | 99.7 | 1.02 | 1.08 | 1.08 | 1.02 | 1.21 | 1.21 |
| CIFAR-10, ResNet34 | 1.802 | 95.3 | 99.8 | 1.01 | 1.06 | 1.19 | 1.01 | 1.06 | 1.19 |

*Table 9.* **Coverage Metrics.** MarCovGap and TopCovGap metrics for dataset-model pairs using LAC, APS, and RAPS algorithms with $\alpha = 0.05$, CP set size 10%, pre- and post-TS calibration.

| Dataset-Model | MarCovGap(%) | | | MarCovGap TS(%) | | | TopCovGap(%) | | | TopCovGap TS(%) | | |
|---|---|---|---|---|---|---|---|---|---|---|---|---|
| | LAC | APS | RAPS | LAC | APS | RAPS | LAC | APS | RAPS | LAC | APS | RAPS |
| ImageNet, ResNet152 | 0.1 | 0 | 0 | 0 | 0 | 0 | 16.1 | 11.5 | 14.3 | 16.5 | 10.1 | 12.4 |
| ImageNet, DenseNet121 | 0 | 0.1 | 0 | 0.1 | 0 | 0 | 15.5 | 12.0 | 15.0 | 16.0 | 11.7 | 14.3 |
| ImageNet, ViT-B/16 | 0.1 | 0 | 0 | 0 | 0.1 | 0.1 | 14.6 | 11.6 | 11.5 | 15.0 | 9.27 | 9.78 |
| CIFAR-100, ResNet50 | 0.1 | 0 | 0 | 0 | 0.1 | 0 | 7.51 | 8.81 | 6.82 | 7.28 | 4.9 | 4.48 |
| CIFAR-100, DenseNet121 | 0 | 0 | 0 | 0 | 0 | 0.1 | 6.72 | 5.91 | 6.50 | 6.50 | 5.41 | 5.41 |
| CIFAR-10, ResNet50 | 0 | 0 | 0 | 0 | 0.1 | 0 | 6.50 | 4.22 | 4.22 | 7.03 | 2.13 | 1.98 |
| CIFAR-10, ResNet34 | 0 | 0 | 0.1 | 0 | 0 | 0 | 4.12 | 2.71 | 2.73 | 4.17 | 1.27 | 1.29 |

see an improvement (lower values), which can be explained by the increase in prediction set sizes. Here, the tendency for the metrics to improve as $T$ increases also remains.

### B.4.1. "MICROSCOPIC" ANALYSIS

In addition to the tables, in order to verify that the increase in the mean set size for APS and RAPS is not caused by a small number of extreme outliers, we "microscopically" analyze the change per sample. Specifically, for each sample in the validation set, we compare the prediction set size after the CP procedure with and without the initial TS calibration (i.e., set size with TS calibration minus set size without). Sorting the differences in a descending order yields a staircase-shaped curve. The smoothed version of this curve, which is obtained after averaging over the 100 trials, is presented in Figure 8 for several dataset-model pairs, both for APS and for RAPS.

For the discussion, let us focus on the CIFAR-100-ResNet50 dataset-model pair in Figure 8. For this pair, we see that approximately one third of the samples experience a negative impact on the prediction set size due to the TS calibration. For about half of the samples, there is no change in set size. Only the remaining small minority of samples experience improvement but to a much lesser extent than the harm observed for others. Interestingly, the existence of samples (though few) where the TS procedure causes a decrease in set size indicates that we cannot make a universal (uniform) statement about the impact of TS on the set size of arbitrary sample, but rather consider a typical/average case.

**ImageNet, ResNet152**   **ImageNet, DenseNet121**   **CIFAR-100, ResNet50**   **CIFAR-100, DenseNet121**

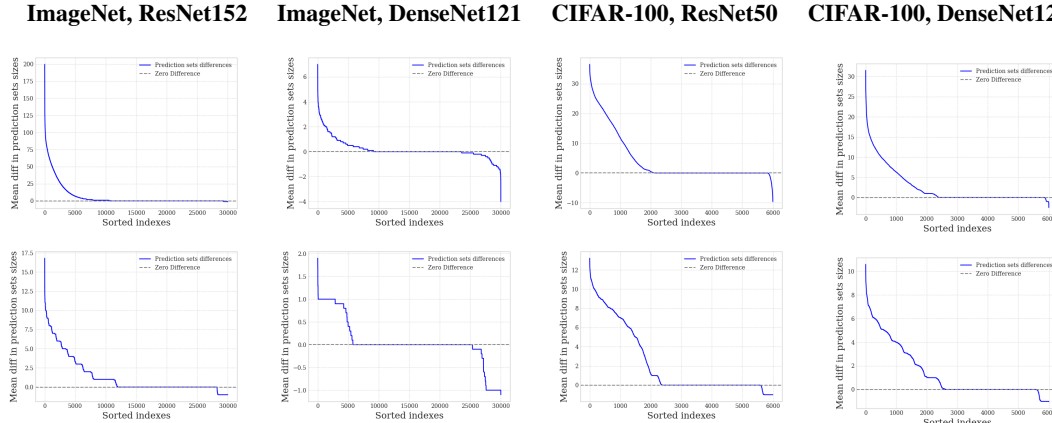

*Figure 8.* Mean sorted differences in prediction set sizes before and after TS calibration for APS (top) and RAPS (bottom) CP algorithms with $\alpha = 0.1$ and CP set size 10%.

## B.5. TS beyond calibration

As an extension of Section 3.3, we provide additional experiments with different settings to examine the effect of TS beyond calibration on CP methods. The figures below present prediction set sizes and conditional coverage metrics for a range of temperatures for additional dataset-model pairs, different CP set sizes and an additional CP coverage probability value. Overall, the temperature $T$ allows to trade off between AvgSize and TopCovGap, as discussed in the paper.

### B.5.1. PREDICTION SET SIZE

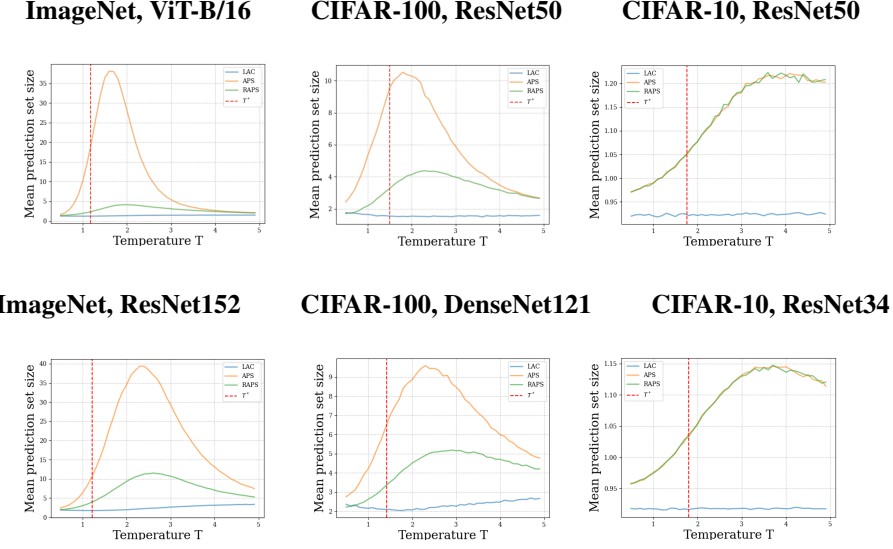

*Figure 9.* **Prediction Set Size.** AvgSize using LAC, APS and RAPS with $\alpha = 0.1$ and CP set size $10\%$ versus the temperature $T$ for additional dataset-model pairs.

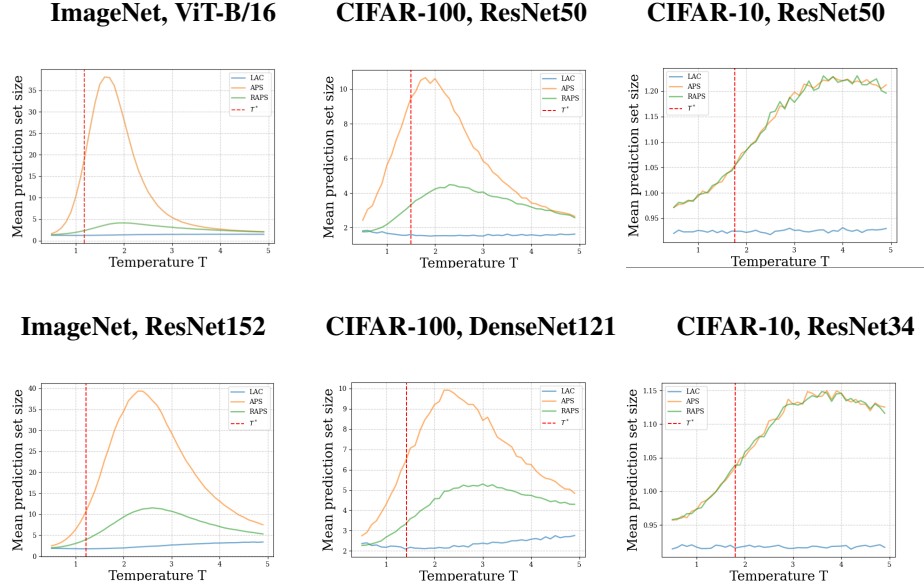

*Figure 10.* **Prediction Set Size.** AvgSize using LAC, APS and RAPS with $\alpha = 0.1$ and CP set size $5\%$, versus the temperature $T$ for additional dataset-model pairs.

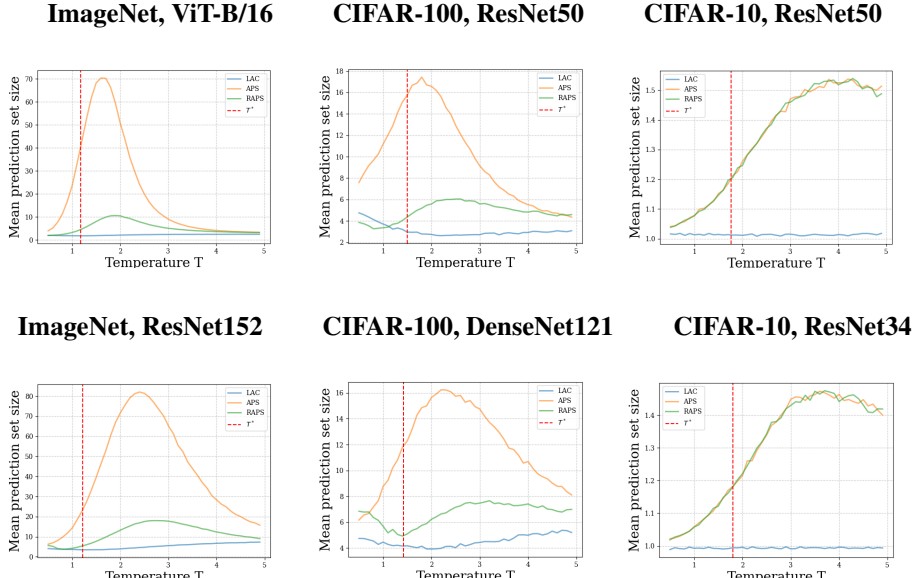

*Figure 11.* **Prediction Set Size.** AvgSize using LAC, APS and RAPS with $\alpha = 0.05$ and CP set size $10\%$ versus the temperature $T$ for additional dataset-model pairs.

### B.5.2. TOPCOVGAP METRIC

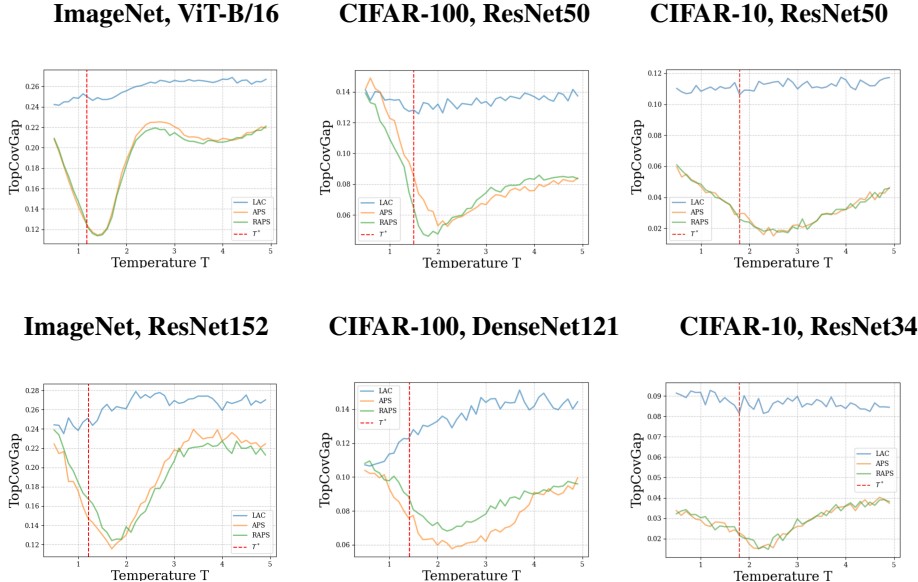

*Figure 12.* **Conditional Coverage Metric.** TopCovGap using LAC, APS and RAPS with $\alpha = 0.1$ and CP set size $10\%$ versus the temperature $T$ for additional dataset-model pairs.

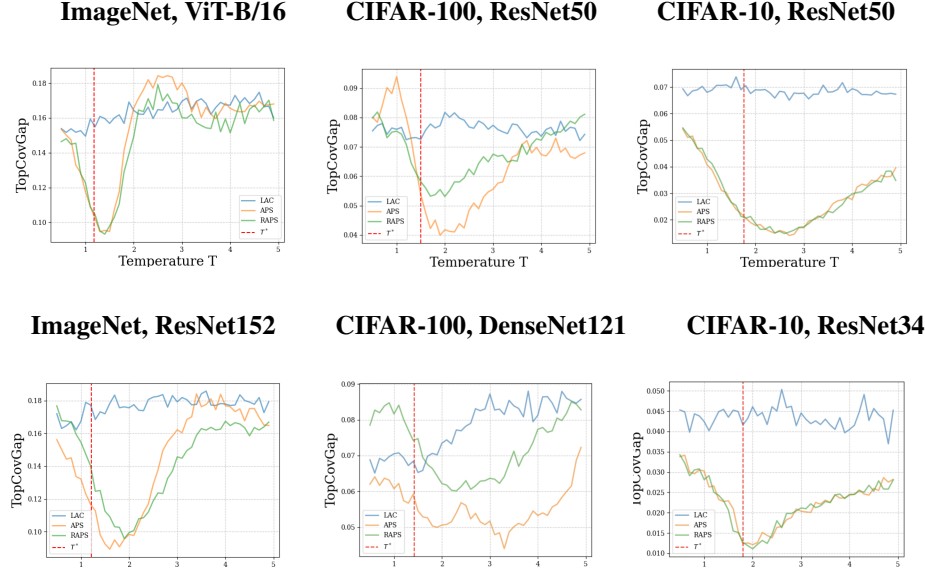

*Figure 13.* **Conditional Coverage Metric.** TopCovGap using LAC, APS and RAPS with $\alpha = 0.05$ and CP set size 10% versus the temperature $T$ for additional dataset-model pairs.

### B.5.3. AVGCOVGAP METRIC

In this subsection, we define and examine the *average class-coverage gap* (AvgCovGap) – a metric that measures the deviation from the target $1 - \alpha$ coverage, averaged across all classes:

$$\text{AvgCovGap} = \frac{1}{C} \sum_{y=1}^{C} \left| \frac{1}{|I_y|} \sum_{i \in I_y} \mathbb{1} \left\{ y_i^{(val)} \in C \left( x_i^{(val)} \right) \right\} - (1 - \alpha) \right|,$$

where $I_y = \{i \in [N_{val}] : y_i^{(val)} = y\}$. The following experimental results for AvgCovGap closely mirror those observed with the TopCovGap metric.

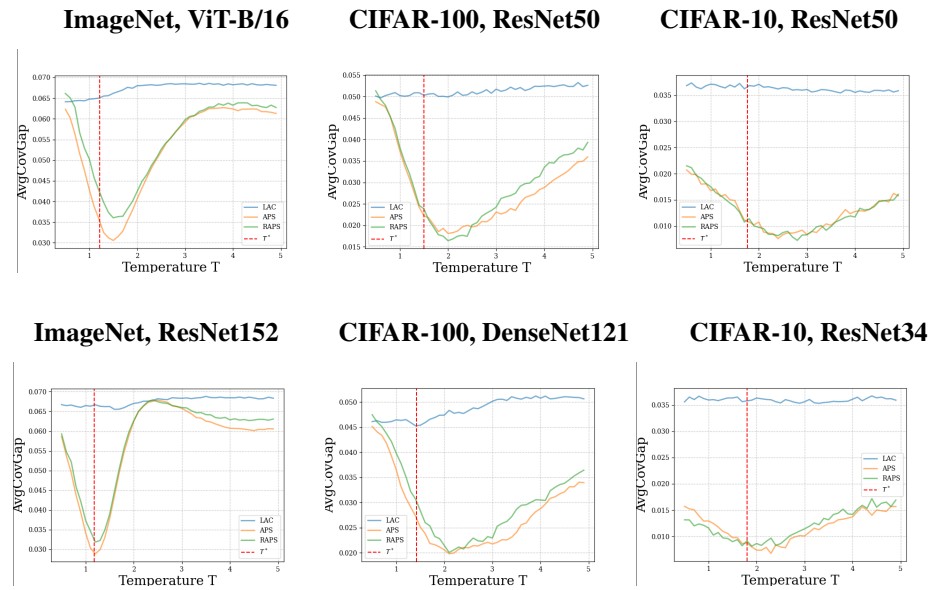

*Figure 14.* **Conditional Coverage Metric**. AvgCovGap using LAC, APS and RAPS with $\alpha = 0.1$ and CP set size 10% versus the temperature $T$ for additional dataset-model pairs.

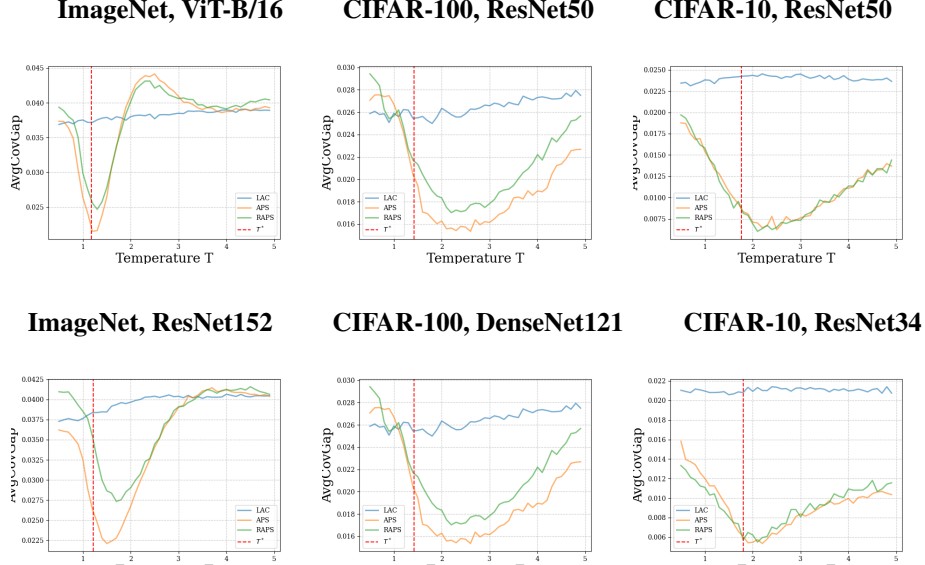

*Figure 15.* **Conditional Coverage Metric**. AvgCovGap using LAC, APS and RAPS with $\alpha = 0.05$ and CP set size 10% versus the temperature $T$ for additional dataset-model pairs.

### B.5.4. CP THRESHOLD VALUE $\hat{q}$

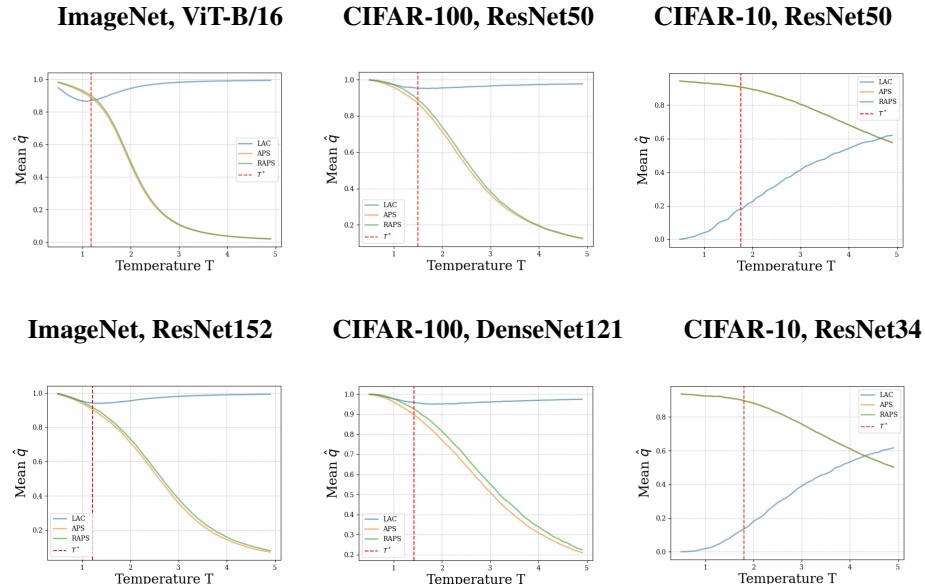

*Figure 16.* **CP threshold value** $\hat{q}$ using LAC, APS and RAPS with $\alpha = 0.1$ and CP set size 10% versus the temperature $T$ for additional dataset-model pairs.

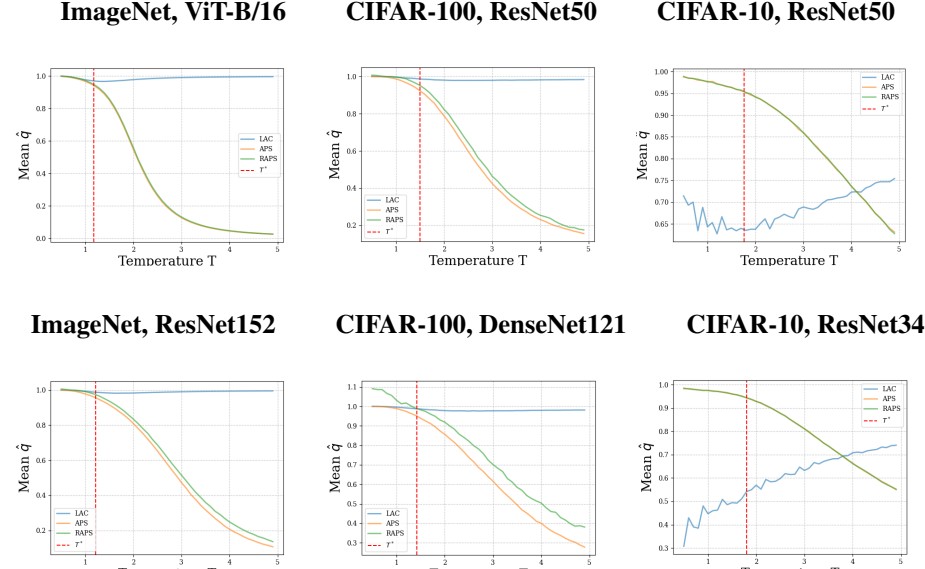

Figure 17. **CP threshold value** $\hat{q}$ using LAC, APS and RAPS with $\alpha = 0.05$ and CP set size $10\%$ versus the temperature $T$ for additional dataset-model pairs.

### B.5.5. THE IMPACT OF TS AT EXTREMELY LOW TEMPERATURES

In our experiments presented in Section 1, we lower bound the temperature range at $T = 0.3$. This choice was motivated by the deviation from the desired marginal coverage guarantee observed at extremely small temperatures. Specifically, we observe that for too small $T$ the threshold value reaches maximal value, $\hat{q} \rightarrow 1$, and, presumably due to numerical errors, this leads to an impractical CP procedure, with significant over-coverage and excessively large prediction set sizes, as demonstrated in Figure 18 for CIFAR100-DenseNet121, $\alpha = 0.1$ and CP set size $10\%$.

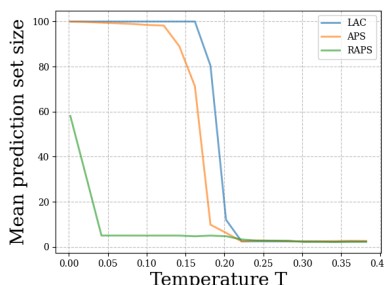

Figure 18. **Prediction Set Size.** Mean prediction set size for LAC, APS and RAPS versus low temperatures, for CIFAR100-DenseNet121.

### B.6. Illustrating the similarity between $\mathbf{z}^q$ and $\mathbf{z}_T^q$

Figure 2 in Section 4.2 shows a strong correlation between the APS score and $\Delta z := z_{(1)} - z_{(2)}$, indicating that quantile samples typically have a dominant entry in their logit vectors ($z_{(1)}^q \gg z_{(2)}^q$). This behavior persists under TS, i.e., for the logits vector $\mathbf{z}_T^q$ of the quantile sample when TS has been applied with some temperature $T$. Recall that the softmax operation applied to the logits vectors is invariant to constant shifts and further amplifies the dominance of larger entries ($\pi_i \propto \exp(z_i)$). Thus, having $z_{T,(1)}^q - z_{T,(2)}^q \approx z_{(1)}^q - z_{(2)}^q \gg 1$, practically implies that $\mathbf{z}_T^q \approx \mathbf{z}^q$.

Here, we demonstrate this similarity using concrete examples of the largest 5 elements of these vector for various dataset–model pairs and temperature settings. To simplify the comparison, we report the values after applying softmax. That is, we report the values of $\boldsymbol{\sigma}(\mathbf{z}_T^q)$.

CIFAR10-ResNet34:

$\boldsymbol{\sigma}(\mathbf{z}^q)[:5] = [9.9997\mathrm{e}{-01}, 8.2661\mathrm{e}{-06}, 7.3281\mathrm{e}{-06}, 6.7146\mathrm{e}{-06}, 2.0070\mathrm{e}{-06}]$

$\boldsymbol{\sigma}(\mathbf{z}_{T=T^*}^q)[:5] = [9.9997\mathrm{e}{-01}, 1.4801\mathrm{e}{-05}, 8.0310\mathrm{e}{-06}, 2.0299\mathrm{e}{-06}, 1.6542\mathrm{e}{-06}]$

$\boldsymbol{\sigma}(\mathbf{z}_{T=0.5}^q)[:5] = [9.9998\mathrm{e}{-01}, 1.1870\mathrm{e}{-05}, 4.1955\mathrm{e}{-06}, 2.0069\mathrm{e}{-06}, 1.9213\mathrm{e}{-06}]$

$\boldsymbol{\sigma}(\mathbf{z}_{T=2}^q)[: 5] = [9.9997\mathrm{e}{-}01, 2.0271\mathrm{e}{-}05, 2.0588\mathrm{e}{-}06, 1.7960\mathrm{e}{-}06, 1.6457\mathrm{e}{-}06]$

CIFAR100-DenseNet121:

$\boldsymbol{\sigma}(\mathbf{z}^q)[: 5] = [9.9993\mathrm{e}{-}01, 5.6212\mathrm{e}{-}05, 5.1748\mathrm{e}{-}06, 3.4063\mathrm{e}{-}06, 1.2153\mathrm{e}{-}06]$

$\boldsymbol{\sigma}(\mathbf{z}_{T=T^*}^q)[: 5] = [9.9992\mathrm{e}{-}01, 4.9569\mathrm{e}{-}05, 8.1525\mathrm{e}{-}06, 4.1283\mathrm{e}{-}06, 1.8294\mathrm{e}{-}06]$

$\boldsymbol{\sigma}(\mathbf{z}_{T=0.5}^q)[: 5] = [9.9997\mathrm{e}{-}01, 1.2354\mathrm{e}{-}06, 6.5692\mathrm{e}{-}07, 4.6121\mathrm{e}{-}07, 4.4120\mathrm{e}{-}07]$

$\boldsymbol{\sigma}(\mathbf{z}_{T=2}^q)[: 5] = [9.9996\mathrm{e}{-}01, 7.2431\mathrm{e}{-}06, 6.0478\mathrm{e}{-}06, 4.3499\mathrm{e}{-}06, 3.7969\mathrm{e}{-}06]$

ImageNet-ResNet152:

$\boldsymbol{\sigma}(\mathbf{z}^q)[: 5] = [9.9991\mathrm{e}{-}01, 5.8738\mathrm{e}{-}05, 1.5782\mathrm{e}{-}05, 8.8207\mathrm{e}{-}06, 4.0592\mathrm{e}{-}06]$

$\boldsymbol{\sigma}(\mathbf{z}_{T=T^*}^q)[: 5] = [9.9992\mathrm{e}{-}01, 3.1237\mathrm{e}{-}05, 9.8604\mathrm{e}{-}06, 9.4194\mathrm{e}{-}06, 3.6732\mathrm{e}{-}06]$

$\boldsymbol{\sigma}(\mathbf{z}_{T=0.5}^q)[: 5] = [9.9993\mathrm{e}{-}01, 3.4423\mathrm{e}{-}05, 2.5229\mathrm{e}{-}05, 1.9685\mathrm{e}{-}05, 1.9673\mathrm{e}{-}05]$

$\boldsymbol{\sigma}(\mathbf{z}_{T=2}^q)[: 5] = [9.9989\mathrm{e}{-}01, 2.9378\mathrm{e}{-}04, 6.1594\mathrm{e}{-}06, 1.5067\mathrm{e}{-}06, 6.1018\mathrm{e}{-}07]$

Across all these dataset–model pairs and temperature settings, the quantile samples exhibit strong similarity. This observation further supports the technical assumption in in Section 4.2 that both $\hat{q}$ and $\hat{q}_T$ correspond to the same underlying sample.

# C. Approximating the Trade-off via the Calibration Set

As discussed in Section 5, exploring the trade-off between prediction set size and class-conditional coverage through the temperature parameter $\hat{T}$ is beneficial when using adaptive CP algorithms.

We propose using TS with two separate temperature parameters on distinct branches: $T^*$, optimized for TS calibration, and $\hat{T}$, which allows for trading prediction set sizes and conditional coverage properties of APS/RAPS to align better with task requirements. One limitation is that the metrics' values for different $\hat{T}$ are not known a priori. However, since we decouple calibration from the CP procedure, the calibration set can be used to evaluate CP algorithms without compromising exchangeability.

The curves in Figure 1 were generated by evaluating the CP methods on the entire validation set (excluding the calibration set and CP set) and averaging over 100 trials. Both are not feasible in practice, where the practitioner only has the calibration set and the CP set of a single trial. Here, we show that these curves can be approximated using only the calibration set for evaluation. In Figure 19, we plot the curves using calibration set + CP set, which together are 20% of the validation set (as in the main body of the paper). Specifically, 10% of the validation set is used for the CP operation (computing the threshold), and the remaining 10% (the original calibration set) serves as a "validation set" to evaluate the CP performance. Therefore, no additional samples are used compared to the common practice of performing calibration and CP calibration sequentially rather than in parallel.

Unlike the curves in Figure 1, the curves in Figure 19 are not averaged over 100 trials but are based on a single trial. Due to randomization, the curves will vary between runs, so to better reflect the practitioner's experience, we present results from 3 separate runs. In each of the runs in Figure 19 the marginal coverage is preserved and the curves of AvgSize and TopCovGap closely resemble the averaged ones shown in Figure 1, demonstrating the user's ability to select $\hat{T}$ based on these single-trial graphs generated using small amount of data. Additionally, note that the procedure required to produce these approximated curves is executed offline during calibration and has a negligible runtime.

*Table 10.* Comparison between MCP vs TS with $\hat{T} = \arg\min_T \text{TopCovGap}(T)$ (enhancing conditional coverage), both based on RAPS, for CIFAR100-ResNet50 and ImageNet-ViT. Recall that lower metrics imply better performance.

| CP set size (%) Metric | CIFAR100-ResNet50 | | | | ImageNet-ViT | | | |
| | MCP | | TS with $\hat{T}$ | | MCP | | TS with $\hat{T}$ | |
| | 10% | 20% | 10% | 20% | 10% | 20% | 10% | 20% |
| --- | --- | --- | --- | --- | --- | --- | --- | --- |
| AvgSize | 4.5 | 8.1 | 2.9 | 3.0 | 7.4 | 6.2 | 2.9 | 3.0 |
| MarCovGap | 0.04 | 0.05 | 0.04 | 0.06 | 0.09 | 0.07 | 0.06 | 0.06 |
| TopCovGap | 0.29 | 0.125 | 0.12 | 0.11 | 0.62 | 0.31 | 0.10 | 0.11 |

## D. Advantages of the Proposed Guidelines over Mondrian Conformal Prediction

In this section, we consider the case of a user that prioritizes class-conditional coverage. In this case, our study recommends applying TS with the temperature of the minimum in the approximated TopCovGap curve, i.e. $\hat{T} = \arg\min_T \text{TopCovGap}(T)$, where TopCovGap is approximated as explained in Appendix C followed by an adaptive CP method like RAPS. Recall that using TS with such temperature can yield high AvgSize, due to the trade-off, but the emphasize here is on prioritizing low TopCovGap.

An existing alternative is to use the Mondrian Conformal Prediction (MCP) approach (Vovk, 2012). MCP aims to construct prediction sets with group-conditional coverage guarantees. Considering the groups to be the classes, the method is based on partitioning the data used for calibration (i.e., the CP set) by classes and obtaining a threshold per class. At deployment, the thresholds are used in a classwise manner. However, a major drawback of MCP is its limited applicability to classification tasks with many classes, since its performance degrades when the number of samples used for calibrating CP per class is small (Ding et al., 2023).

Note that in our experiments, we consider CIFAR-100 that has 100 classes and CP set (used to calibrate the CP) of size up to 2000 (20% of the validation set), and ImageNet that has 1000 classes and CP set of size up to 10000 (20% of the validation set). This means that, approximately, we have up to 20 samples per class to calibrate CP for CIFAR-100 and up to 10 samples per class to calibrate CP for ImageNet.

Table 10 presents the metrics AvgSize, MarCovGap and TopCovGap for our proposed approach and for MCP, when both utilize RAPS, for the dataset-model pairs CIFAR100-ResNet50 and ImageNet-ViT. The results demonstrate the superiority of using TS with $\hat{T} = \arg\min_T \text{TopCovGap}(T)$ (computed based on approximated curve) over MCP across all metrics. Recall that we consider the case where class-conditional coverage is preferred, and indeed, TS with such temperature constantly yields better TopCovGap than MCP. Yet, interestingly, it outperforms MCP also at AvgSize.

To conclude, the experiments presented in this section further shows the practical significance of our guidelines.

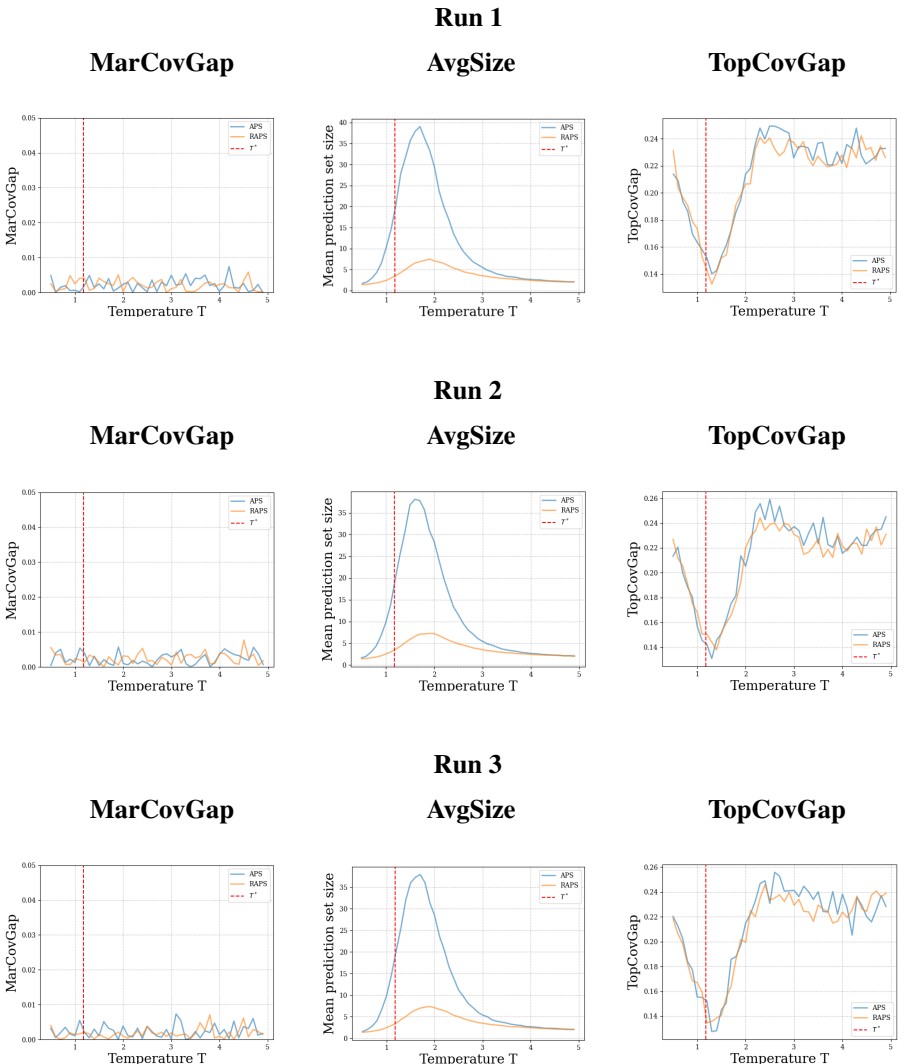

*Figure 19.* **Performance evaluation with small data.** Examining the performance of APS and RAPS with small evaluation data for ImageNet-ViT-B/16 with $\alpha = 0.1$ and CP set size 10%. Each row displays the marginal coverage, prediction size and conditional coverage metrics that are computed over 1 trial using 10% of the validation set.

