# OpenReview forum: "On Temperature Scaling and Conformal Prediction of Deep Classifiers"
_ICML.cc/2025/Conference — ICML 2025 poster_

### Official Review · Reviewer_paq4 · 2025-03-05

**Overall Recommendation:** 3

**Summary:**

This paper focuses on a popular calibration technique known as temperature scaling (TS) and investigates its effect on major conformal prediction (CP) methods (LAC, APS, RAPS). They show that TS improves class-conditional coverage of adaptive CP but increases prediction set sizes; the effect on LAC is negligible. They uncover a trade-off between prediction set size and conditional coverage when modifying the temperature in TS, with some theoretical analysis.

**Claims And Evidence:**

* Empirical
  * The paper claims to have an extensive empirical study on DNN classifiers to demonstrate how TS affects CP. Numerical experiments are conducted on three CP methods over a range of classification dataset. Several findings are summarized from the experiments, and the appendix contain more numerical details regarding hyperparameter settings and furture comparison. The experiments are pretty thorough and well documented.
* Theoretical analysis
  * Compared to the empirical part, the theoretical side is a bit straightforward but is compatible with the main finding, in part. It does not concern the conditional coverage, but mainly focus on the prediction set size. Regarding the non-monotone structure, Proposition 4.3 does have some theory regarding T=1, but this is more indirect rather than the true kink point in the non-monotone relationship.

**Essential References Not Discussed:**

N/A

**Experimental Designs Or Analyses:**

The experimental design and analyses appear to be sounds. The authors use relevant datasets, metrics, and comprehensive comparison. The theoretical analysis provides insights into the empirical observations. Limitations and future directions are appropriately discussed.

**Methods And Evaluation Criteria:**

* The evaluation metrics of conditional coverage (impossible in general) and prediction set size are very well recognized in conformal prediction literature.

* The numerical experiments are conducted in a range of data sets.

**Other Comments Or Suggestions:**

N/A

**Other Strengths And Weaknesses:**

N/A

**Questions For Authors:**

N/A

**Relation To Broader Scientific Literature:**

Conformal prediction provides guarantee for high-risk decision making and warrants further development. This paper's finding helps to refine calibration technique of CP.

**Theoretical Claims:**

* The theoretical claims seem intuitive and correct but I only had limited review of the proofs. On the other hand, the result is not strong and more auxiliary to support the empirical findings.

---

> ### Author Rebuttal · Authors · 2025-03-31
>
> Thank you for your insightful review. We appreciate your recognition of our extensive empirical study and clear experimental design.
>
> Below, we address your comments on the theoretical analysis point by point.
>
> ***
>
> > Theory for conditional coverage.
>
> ***
>
> Our work develops a comprehensive mathematical framework to explain the non-intuitive effects of TS on prediction set sizes in CP. We rigorously formulate conditions for changes in prediction set size and clarify the underlying theoretical reasons for these effects. While extending this analysis to conditional coverage is valuable, it would require an entirely new theoretical framework with different assumptions and tools, making it beyond the scope of this study. Nonetheless, establishing theory for the impact of TS on conditional coverage is indeed a promising direction for future research.
>
> ***
>
> > Regarding Proposition 4.3.
>
> ***
>
> Let us clarify the scope of our theoretical analysis and particularly the contribution of Proposition 4.3. This proposition focuses on how TS affects prediction set size of APS. It provides a condition, dependent on the value of $ T $, for a decrease or an increase in the prediction set size after applying TS, denoted by $L_T$, compared to the size without applying TS (equivalently, TS with $T=1$), denoted by $L$. It is important to note that this result covers all positive values $ T > 0 $.
>
>
> We would like to clarify that this is not the primary finding of our work. Theorem 4.4, along with its implications, specifically addresses this "kink point" of the non-monotone dependency. Theorem 4.4 shows that applying TS with temperature $T$ affects the prediction set size of a sample $z$ based on $ \Delta z = z_{(1)} - z_{(2)}$ and a bound $ b(T) $. For example, for $ T > 1 $, if $ \Delta z > b_{T>1}(T) $, the prediction set size increases ($ L_T \geq L $). Figure 3 reveals a non-monotonic curve for $ b(T) $ at $T>1$, and since lower (resp. higher) $ b(T) $ implies that more (resp. less) samples obey the condition this provides an explanation to the non-monotonic pattern of the mean prediction set size (AvgSize): it increases for $ 1 < T < \tilde{T}_c $ and decreases for $ T > \tilde{T}_c $ where $\tilde{T}_c$ is the temperature above 1 where $ b(T) $ is minimal. This critical temperature $ \tilde{T}_c $ decreases with $ C $, aligning with empirical results.

---

> > ### Comment · Reviewer_paq4 · 2025-04-05
> >
> > Thanks for the helpful response. I am maintaining my score based on my understanding.

---

### Official Review · Reviewer_t6t7 · 2025-03-14

**Overall Recommendation:** 3

**Summary:**

In this work, the authors studied the effect of the widely used temperature scaling calibration on the performance of conformal prediction techniques for deep neural network classifiers. A wide range of experiments are conducted and a theoretical framework is proposed to explain the effects.

**Claims And Evidence:**

The authors have provided a theoretical analysis to show how temperature values influence the properties of prediction sets. With the theoretical results, researchers can understand why temperature scaling can affect the conformal prediction. Empirically, the experiments are strong and extensive.

**Essential References Not Discussed:**

Seems not applicable.

**Experimental Designs Or Analyses:**

No concerns are raised.

**Methods And Evaluation Criteria:**

The experimental settings are strong, covering various datasets, neural network backbones, etc.  The descriptions of the validations are clear.

**Other Comments Or Suggestions:**

Some suggestions:

1. It would increase the readability if the font size and line width in the figures are increased.

2. Providing clearer insights/explanations to link some proposed theories and their practical values would further enhance the practicality or be attractive for the practitioners in the ML community.

Additional questions:

Can the authors provide some insights on the applicability of the findings in other tasks not limited to image classification?

**Other Strengths And Weaknesses:**

Additional strength: Basically, the paper is well written and organized. The appendices are extensive.

**Questions For Authors:**

No further comments.

**Relation To Broader Scientific Literature:**

Broad literature, including the recent achievements, are duly included in the paper.

**Theoretical Claims:**

Because this work falls beyond my research comfort zone, a serious assessment of the proposed theories in detail is a challenging task for me. Therefore, this innitial evaluation may be conservative. I am eager to actively follow the discussions with the authors, other reviewers, and ACs.

---

> ### Author Rebuttal · Authors · 2025-03-31
>
> Thank you for your thoughtful review and valuable suggestions. We appreciate your recognition of our comprehensive experiments and clear presentation.
>
> Below, we provide a point-by-point response to your comments.
>
> ***
>
> > Improving readability of the figures.
>
> ***
>
> Following this comment, in the revision we will make every effort to improve the readability of the figures. The extra page allowed in the final version will provide additional space to further enhance readability.
>
> ***
>
> > Practicality of the proposed theory.
>
> ***
>
> The primary aim of our theoretical study is to provide mathematical reasoning for the surprising empirical behavior that we discovered. We believe that addressing the "why" question behind non-intuitive results is of high scientific importance.
>
> That being said, our theory also provides practical implications. For example, if a practitioner cares only about AvgSize and wants to use APS/RAPS, the $0<T<1$ branch of our theory suggests that they pick small $T$ (which is aligned with the empirical observations).
> Conversely, practitioners can leverage the $T>1$ branch of our theory, where monotonicity in CP properties breaks. Specifically, it justifies defining a finite range for tuning $T$ to balance prediction set sizes and class-conditional coverage, as discussed in Section 5.    We refer the reviewer to Appendix C, where we demonstrate how a small calibration dataset can approximate the curves for AvgSize and TopCovGap versus $T$ (shown in Figure 1). Based on these approximate trends, users can select an optimal $\hat{T}$ that aligns with their needs. Additionally, in Appendix D, we highlight the advantages of this approach over existing methods.
>
> ***
>
> > Applicability to other domains.
>
> ***
>
> Our work deals with temperature scaling and conformal prediction applied to multiclass classification.
> We used image classification datasets (ImageNet, CIFAR-100, CIFAR-10) as they are the benchmark datasets in the related literature.
> Nevertheless, we expect our findings to be beneficial to classification tasks in domains other than images.
>
> In particular, the practical relevance of our paper lies in empowering users to effectively apply CP in multiclass classification, based on their specific needs, by following our proposed guidelines. The applications extend to domains other than images where CP has been used, such as medical diagnosis [1],[2] and NLP [3],[4].
>
> [1] Lu, Charles, et al (2021). "Fair conformal predictors for applications in medical imaging." Proceedings of the AAAI Conference on Artificial Intelligence.
>
> [2] Vazquez, J., Facelli, J. C. (2022). "Conformal Prediction in Clinical Medical Sciences." Journal of healthcare informatics research,
>
> [3] Kumar, Bhawesh, et al (2023). "Conformal prediction with large language models for multi-choice question answering." ICML 2023.
>
> [4] Campos, Margarida, et al (2024). "Conformal prediction for natural language processing: A survey."

---

### Official Review · Reviewer_8jYt · 2025-03-14

**Overall Recommendation:** 4

**Summary:**

Calibration and Conformal Prediction (CP) are two popular approaches to solving the overconfidence problem in modern DNN classifiers. In this paper, the authors studied the effects of temperature scaling (TS), which is effective for calibration, on the efficiency of CP. The authors first designed extensive experiments to show how the TS affects the sizes of the prediction sets and class-conditional coverage. Then, the authors conducted theoretical analyses for the empirical observations and finally provided practical guidelines for choosing the temperature parameter.

## update after rebuttal
The authors have addressed my concerns, so I would support acceptance.

**Claims And Evidence:**

The main claim made in this work is that TS has little effect on LAC but strongly affects the prediction set's size and conditional coverage of APS and RAPS. Besides, the effects of the temperature parameter are non-monotonic. These claims were empirically observed and then were analyzed theoretically.

**Essential References Not Discussed:**

The related references are properly cited and discussed.

**Experimental Designs Or Analyses:**

The experiments in Section 3 are sufficient to support the authors’ empirical analysis of the effect of TS on CP. The experiments in Section 4 and the appendix are also enough to verify the authors' claims.

**Methods And Evaluation Criteria:**

The evaluation criteria used in the empirical studies, including marginal coverage gap, average set size, and top-5% class coverage gap, are meaningful.

**Other Comments Or Suggestions:**

In Line 1087, $d$ in the LHS of the inequality should be $g$.

**Other Strengths And Weaknesses:**

Strength:

- The effect of TS on the performance of CP is discussed for the first time.

- The empirical studies are extensive.

- Theoretical analyses are provided.

Weakness:

- Some details of the theoretical analyses should be refined.

**Questions For Authors:**

My questions are provided in the above comments.

**Relation To Broader Scientific Literature:**

The prior related studies only use initial TS calibration before applying CP methods [1, 2, 3, 4]. None of them explored the effect of TS on CP, which is analyzed in detail in this paper.

[1] Angelopoulos, A. N., Bates, S., Jordan, M., and Malik, J. Uncertainty sets for image classifiers using conformal prediction. In International Conference on Learning Representations, 2020.

[2] Lu, C., Yu, Y., Karimireddy, S. P., Jordan, M., and Raskar, R. Federated conformal predictors for distributed uncertainty quantification. In International Conference on Machine Learning, pp. 22942–22964. PMLR, 2023.

[3] Gibbs, I., Cherian, J. J., and Cand` es, E. J. Conformal prediction with conditional guarantees. arXiv preprint arXiv:2305.12616, 2023.

[4] Lu, C., Yu, Y., Karimireddy, S. P., Jordan, M., and Raskar, R. Federated conformal predictors for distributed uncertainty quantification. In International Conference on Machine Learning, pp. 22942–22964. PMLR, 2023.

**Theoretical Claims:**

The overall theoretical claims seem correct, but some details should be corrected or might be more clearly clarified.

- The proof for the Lemma for Theorem 4.1(A.1) seems not correct, or more specifically, Line 575 to 578 is not reasonable. In fact, this lemma can be more simply proved by investigating the monotonicity of function $\exp\left(\frac{z_i-z_j}{t}\right)$ with respect to $t$.

- According to the proof for Proposition 4.3 in Appendix A, it seems that, in Line 322 ‘M∈[L]' should be changed to ‘M∈[L_T]' if 0<T<1, and in Line 325 ‘M∈[L_T]' should be changed to ‘M∈[L]' if T>1. Besides, summation signs are missing in Lines 322 and 325, comparing Proposition 4.3 and A.3, which are indeed the same proposition.

- Could the authors give more explanations for $\pi^q\approx\pi^q_T$ in Line 353?

- In the proof of Theorem 4.4, should $\frac{T-1}{T+1}\ln\frac{4(C-1)^2}{T}$ be changed to $\frac{T}{T+1}\ln\frac{4(C-1)^2}{T}$? And consequently the results in the theorem.

I suggest that the authors should carefully check all the details of the theoretical parts.

---

> ### Author Rebuttal · Authors · 2025-03-31
>
> We are grateful for your constructive  and thorough review of our theoretical work. We appreciate your acknowledgment of our novel analysis, supported by extensive empirical studies and theoretical investigation. We have carefully addressed your comments and suggestions.
>
> Below, we provide a point-by-point response to your comments.
>
> ***
>
> > Regarding the proof for Lemma for Theorem 4.1 (A.1)
>
> ***
>
> Thank you for highlighting this issue. You are correct — there was an error and a simple proof will be constructed in the revision based on the monotonic decrease of $\exp(c/t)$ with respect to $t$ when $c>0$. Indeed,
>
> $\\exp(z_{i}/\\tilde{T}) \\cdot \\exp(z_{j}/T) \\geq \\exp(z_{i}/T) \\cdot \\exp(z_{j}/\\tilde{T}) \iff \\exp((z_{i}-z_j)/\\tilde{T}) \\geq \\exp((z_{i}-z_j)/T)$
>
> and since $z_{i}-z_j \geq 0$ we have that this inequality holds as $\tilde{T} \leq T$.
>
> ***
>
> > Typo in Proposition 4.3
>
> ***
>
> Thank you for pointing out this typo.
>     Indeed, in the branch $0<T<1$,  $M\in\left[L\right]$ should be changed to $M\in\left[L_T\right]$, and in the branch $T>1$, $M\in\left[L_T\right]$ should be changed to $M\in\left[L\right]$.
>     Similarly, in the proof, in lines 685 and 688, $L$ should be replaced with $L_T$ (3 instances).
>
> As for the sum symbol, we intended to save space by using
> $\\sum_{i}^{M} \\hat{\\pi}_i - \\hat{\\pi}^T_i $, which we read as $\sum_i^{M} \left( \hat{\pi}_i - \hat{\pi}^T_i \right)$, rather than $\\sum_i^M \\hat{\\pi}_i - \\sum_i^M \\hat{\\pi}_i^T$.
>
> In the revision, we will ensure consistent phrasing in the main body of the paper and the appendix.
>
> ***
>
> > Explanation for $\pi^q \approx \pi^q_T$ in Line 353.
>
> ***
>
> The paragraph in lines 332-357 analyzes the structure of the "quantile sample" of APS. Figure 2 shows a strong correlation between the samples' scores and $\Delta z = z_{(1)} - z_{(2)}$ (the difference between the largest and second largest entries of the logits vector), both with and without temperature scaling. Since the quantile sample corresponds to a high score ($1 - \alpha$ quantile of all scores), it consistently has a large $\Delta z$. As evidenced in Figure 2, the $1 - \alpha = 0.9$ quantile sample had $\Delta z^q \approx 11$. According to the relation $\pi_i \propto \exp{z_i}$, this implies $\pi^q_{(1)} \gg \pi^q_{(2)}$, specifically, after some approximations we get $\frac{\pi^q_{(1)}}{\pi^q_{(2)}} \propto 10^4$. Similarly, when temperature scaling calibration was applied,  the $1 - \alpha = 0.9$ quantile sample again resulted in $\Delta z_T^q \approx 11$, leading to $\pi_{T,(1)}^{q} \gg \pi_{T,(2)}^{q}$. Thus, due to the overwhelmingly dominant entry in both quantile samples, we have that $\pi_{(1)}^{q}$ and $\pi_{T,(1)}^{q}$ are nearly 1, and the associated sorted softmax vectors obey $\pi^q \approx \pi^q_T$.
>
> Following this comment, to further illustrate the strong similarity between sorted $\pi^{q}$ and $\pi^{q}_T$, we will add in the revised version concrete examples for these vectors for several dataset-model pairs. For example, the first 5 elements in $\pi^{q}$ and $\pi^{q}_T$ (after sorting) for ImageNet-ViT (with $T = T^*$ optimal for calibration):
>
> $\pi^q[:5] = [9.9697e-01, 4.9439e-04, 2.1028e-04, 2.0065e-04, 1.6687e-04]$
>
>
> $ \pi^{q}_T[:5] = [9.9755e-01, 7.6261e-04, 8.7435e-05, 7.8683e-05, 7.6373e-05]$
>
> ***
>
> > Regarding the bound in Theorem 4.4.
>
> ***
>
> Thank you for pointing out this typo. Indeed, there is a factor of $\frac{T}{T+1}$ in the $T>1$ branch of Theorem 4.4 and not $\frac{T-1}{T+1}$.
> The same correction applies to lines 787 and 828 in the proof.
>
> We emphasize that the proof remains valid except for the final line of each branch (787, 828), where $T-1$ was mistakenly written instead of $T$ in the numerator of the bound. This will be fixed.
> After this minor correction, the bounds are still aligned with the empirical trends.
>
> The complete substitution of $A$ from line 756 into the inequality in line 782, detailed in the following link: https://postimg.cc/hXXHxKNq confirms the correctness of $\max \left( \frac{T}{T-1}\ln(4T),\frac{T}{T+1}\ln(4T(C-1)^2) \right)$ for the branch $T>1$.
>
> Following this comment, we will include this substitution explicitly in the revised version.

---

### Official Review · Reviewer_doKf · 2025-03-21

**Overall Recommendation:** 2

**Summary:**

The paper aims to study the interplay between conformal prediction (CP) and temperature scaling (TS) calibration. They study the effect of TS on conformal prediction using extensive empirical evaluation with three different CP methods. They present a theoretical analysis to explain the effect of TS on APS and RAPS conformal prediction methods.

**Claims And Evidence:**

The paper presents theory to support their claims. I am unsure about the assumption that $\hat{q}$ and $\hat{q}_T$ correspond to the same sample based on Figure 2. How generalizable is this claim? There can always be adversarial sequences that violate this, which puts the assumption in question. If not justifiable theoretically, at the very least more extensive empirical analysis to demonstrate this holds for datasets included in the paper and beyond is important.

**Essential References Not Discussed:**

While the paper claims this interplay has not been investigated yet, the paper misses an important reference [1] that studies this very impact of confidence calibration on conformal prediction. From a first glance, the analysis and empirical observations of [1] are close to this work. The paper also does not cite [2] who study the connection between calibration and prediction sets, although in the binary classification setting. These are just a few examples and not an exhaustive list! Authors should acknowledge these works and include a discussion on relationship with these works at the very least. The authors are also suggested to do a thorough review of existing literature to contextualize their work better.

[1] Xi, H., Huang, J., Liu, K., Feng, L., and Wei, H. (2024). Does confidence calibration improve conformal prediction?

[2] Gupta, C., Podkopaev, A., and Ramdas, A. (2020). Distribution-free binary classification: prediction sets, confidence intervals and calibration. NeurIPS.

**Experimental Designs Or Analyses:**

I checked the soundness of experiments in the main paper.

**Methods And Evaluation Criteria:**

While the paper presents extensive empirical evaluation and the datasets make sense, I am concerned about the metrics included. The justification behind evaluating class-coverage gap on top-5% classes is not provided; moreover, 5% seems an arbitrary choice. Why would you not consider evaluation on average class coverage gap as defined in Ding et al., 2023? At the very least, both metrics could be included. Also, ablation on the 5% choice seems important to make general claims about class-conditional coverage. My other concern is regarding absence of standard error reporting in the experiments.

**Other Comments Or Suggestions:**

The definition for reported metrics (pg 4) should be included in the main paper for improved readability.

**Other Strengths And Weaknesses:**

The writing of the paper, especially the technical writing and notation in theorems can be improved for greater clarity. I am also unsure about the practical utility of the findings presented – temperature scaling is not a core component of conformal methods and can be done away with. While the authors present some guidelines, it does not seem convincing in the current context. Additionally, the authors mention the runtime of their procedure (pg 8). I have two comments here – (i) please discuss the runtime of your procedure in the paper, (ii) I believe offline training of DNNs should not be compared here given the post-hoc nature of methods.

**Questions For Authors:**

No specific questions, please refer to individual comments above.

**Relation To Broader Scientific Literature:**

The paper aims to study the interplay between conformal prediction and temperature scaling calibration – methods that have been often studied individually in literature. TS is usually applied initially in conformal prediction methods. They report their findings on the effect of TS on conformal prediction sets in terms of set size and class-conditional coverage.

**Theoretical Claims:**

I went over proofs A.1-A.4.

---

> ### Author Rebuttal · Authors · 2025-03-31
>
> Thank you for your thorough and insightful review. We are pleased that you recognized the extensive empirical evaluation of our experimental design.
>
> Below, we carefully respond to all of your comments.
>
> > Regarding the technical assumption
>
> The core of this technical assumption is the strong similarity between the softmax vectors of the "quantile samples" associated with $\hat{q}$ and $\hat{q}_T$, as discussed in the paragraph in lines 332-357.
>
> Due to the limitation of characters, we refer you to the response "Explanation for $\pi^q \approx \pi^q_T$ in Line $353$" to Reviewer 8jYt, where we explain and illustrate the proximity between $\pi^q $ and $ \pi^q_T$. To highlight the strong similarity, in the revised version we will include concrete examples of these vectors for various dataset-model pairs.
>
> Given that $\pi^q \approx \pi^q_T$, we believe that it is reasonable to make the theoretical derivation tractable by the technical assumption that $\hat{q}$ and $\hat{q}_T$ correspond to the same sample. We do not claim that it holds in every possible setting. Nevertheless, the fact that our theory provides insights that are aligned with the empirical behavior along many models and datasets serves as a justification for this technical assumption.
>
> > Metrics for class-conditional coverage
>
> Evaluating conditional coverage across groups using the worst-case coverage metric is both informative and relevant, as demonstrated by previous works, e.g., (Gibbs et al., 2023). Our TopCovGap metric is averaged over the worst 5\% due to high variance observed when considering only the single worst-case coverage, as explained in Appendix B.4.
>
> Following your comment, we will report also the AvgCovGap metric that has been used in (Ding et al., 2023) (denoted CovGap there). For example, we present in https://postimg.cc/1fCkf8CH this metric for the three dataset-model pairs shown in Figure 1. Notably, this metric exhibits similar behavior to TopCovGap, displaying a comparable trend of achieving a minimum at temperatures $T>1$.
>
> > Missing references
>
> We thank you for bringing these related papers to our attention. Both papers will be cited and discussed in the revised version.
>
> Paper [1] (arXiv preprint) is a concurrent work (our paper was uploaded to arXiv at the same time).
> The TS results in [1] are only a small subset of our results. They do not consider conditional coverage and use a limited range of T, which masks the non-monotonic effect on the prediction set size of APS and RAPS. Moreover, they also do not compare APS and RAPS to LAC. On the other hand, our paper provides a complete picture of the effect of TS with a wide range of temperatures on both the prediction set size and the class-conditional coverage of APS, RAPS, and LAC. This complete picture teaches practitioners that tuning the temperature for APS and RAPS introduces a trade-off effect (overlooked in [1]), and we provide a practical way to control it.
>
>
> Paper [2] presents important theoretical results on prediction sets and calibration, which are indeed related to our work.
> Note though, that [2] is limited to binary classification and do not provide explanation to the fact that calibration (e.g., TS calibration) affects CP methods differently and in a non-monotonic way, as empirically shown and analyzed in our paper.
>
> > Practical utility of the findings
>
> Our work investigates the effect of TS on CP as a function of temperature $T$. Through our analysis, we identify a trade-off between two essential properties of adaptive CP methods: mean prediction set size and proximity to conditional coverage. Our guidelines, introduced in Section 5, enable practitioners to navigate this trade-off effectively, which is a novel contribution to the CP literature. The guidelines are further explored in Appendix C and D.
>
> Appendix C demonstrates how, with a limited amount of calibration data, users can approximate the curves in Figure 1. Based on these trends, they can select an appropriate $\hat{T}$ that aligns best with their objectives. For instance, a practitioner working with CIFAR100-ResNet50 who prioritizes prediction set size can achieve over a 50\% reduction in the AvgSize of APS using our guidelines.
>
> Appendix D further illustrates the practical relevance of our findings. Specifically, we show that applying TS at the temperature corresponding to the minimum of the *approximated* TopCovGap curve, followed by RAPS, outperforms Mondrian CP (Vovk, 2012) in both TopCovGap and AvgSize metrics.
>
> > Regarding the runtime
>
> In the revised version we will discuss more about the runtime of the proposed guidance. We would like to emphasize that this procedure is done offline during the calibration phase and its runtime is within a range of minutes.
>
> > Improving readability
>
> We will include the definitions of the reported metrics (currently appear in the appendix) in the main body of the revised version leveraging the extra page allowed in the final version.

---

> > ### Comment · Reviewer_doKf · 2025-04-06
> >
> > Thank you for your response.
> >
> > **Technical assumption:** I saw the response and I believe you continue to point to 332-357 and Figure 2. I do not believe it is fair to make a broad assumption based on one dataset and model. While I understand you will add more examples in the revision, in the current form, the justification is not satisfactory. "We do not claim that it holds in every possible setting." -- what are the settings where you claim this holds? This discussion is entirely missing. It is a strong assumption and if it is believed to hold for some specific datasets and models, it should be proved. If it is expected to hold only for the datasets and models you show the trend for, that should also be mentioned clearly. This affects the generalizability of the findings and seems important to me.
> >
> > **Metrics for class-conditional coverage:** I believe Gibbs et al. used worst-case coverage only at one place and mention that a practitioner may choose to prioritize different conditional targets. Most of the experiments report miscoverage over all groups.
> >
> > **runtime:** I believed the discussion of runtime is important since you compare the method with standard conformal prediction that has no such overhead. My comment was with respect to this line in the paper -- "negligible runtime compared to the offline training of DNNs" -- it is not fair to state this since post-hoc conformal methods assume access to pretrained model.
> >
> > I am still concerned about the reported metrics and lack of error bars among other things, and I would like to keep my score.

---

> > > ### Author Response · Authors · 2025-04-08
> > >
> > > Thank you for your comment. We are glad for the opportunity to further dive into your concerns and resolve them.
> > >
> > > &nbsp;
> > >
> > > ### Technical assumption
> > >
> > > We emphasize that we did not claim that $\hat{q}$ and $\hat{q}_T$ are associated with exactly the same sample in any specific setting. Rather, we motivate this technical assumption, which makes the theoretical derivation tractable, by showing that the associated softmax vectors $\pi^q$ and $\pi^q_T$ are aligned. Figure 2 illustrates this for a specific dataset-model pair, and this behavior holds across **all other dataset-model pairs** as well. We previously provided concrete examples of softmax vectors for ImageNet-ViT, and in this comment, we include additional dataset-model pairs along with different values of temperatures for further illustration.
> > > As mentioned, we will include these additional demonstrations for all other dataset-model pairs we experimented with in the revised version.
> > >
> > > Moreover, please note that in the CP literature, assumptions are often necessary when developing rigorous theories beyond marginal coverage. For example, as mentioned in Section 2.2, APS (Romano et al., 2020) and LAC (Sadinle et al., 2019) establish theory under the assumption that the classifier outputs the exact posterior distribution— stronger assumption than ours, and without empirical support.
> > >
> > > Finally, as previously noted, our theory aligns well with empirical observations across models and datasets, which further provides  justification for the technical assumption.
> > >
> > >
> > > #### CIFAR10-ResNet34:
> > >
> > > $\pi^q = [9.9997e-01, 8.2661e-06, 7.3281e-06, 6.7146e-06, 2.0070e-06]$
> > >
> > > $\pi^q_{T = T^*} = [9.9997e-01, 1.4801e-05, 8.0310e-06, 2.0299e-06, 1.6542e-06]$
> > >
> > > $\pi^q_{T = 0.5} = [9.9998e-01, 1.1870e-05, 4.1955e-06, 2.0069e-06, 1.9213e-06]$
> > >
> > > $\pi^q_{T = 2} = [9.9997e-01, 2.0271e-05, 2.0588e-06, 1.7960e-06, 1.6457e-06]$
> > >
> > > #### CIFAR100-DenseNet121:
> > >
> > > $\pi^q = [9.9993e-01, 5.6212e-05, 5.1748e-06, 3.4063e-06, 1.2153e-06]$
> > >
> > > $\pi^q_{T = T^*} = [9.9992e-01, 4.9569e-05, 8.1525e-06, 4.1283e-06, 1.8294e-06]$
> > >
> > > $\pi^q_{T = 0.5} = [9.9997e-01, 1.2354e-06, 6.5692e-07, 4.6121e-07, 4.4120e-07]$
> > >
> > > $\pi^q_{T = 2} = [9.9996e-01, 7.2431e-06, 6.0478e-06, 4.3499e-06, 3.7969e-06]$
> > >
> > > #### ImageNet-ResNet152:
> > >
> > > $\pi^q = [9.9991e-01, 5.8738e-05, 1.5782e-05, 8.8207e-06, 4.0592e-06]$
> > >
> > > $\pi^q_{T = T^*} = [9.9992e-01, 3.1237e-05, 9.8604e-06, 9.4194e-06, 3.6732e-06]$
> > >
> > > $\pi^q_{T = 0.5} = [9.9993e-01, 3.4423e-05, 2.5229e-05, 1.9685e-05, 1.9673e-05]$
> > >
> > > $\pi^q_{T = 2} = [9.9989e-01, 2.9378e-04, 6.1594e-06, 1.5067e-06, 6.1018e-07]$
> > >
> > > &nbsp;
> > >
> > > ### Metrics for class-conditional coverage
> > >
> > > Please note that all the other reviewers supported our TopCovGap metric.
> > > We recognize the value of reporting AvgCovGap (averaged across all groups, as used in (Gibbs et al., 2023)) alongside TopCovGap. As shown in our previous comment in  the following **link**: https://postimg.cc/1fCkf8CH, **the observed trends are very similar**. As we stated in the previous response, in the revised version, we will report AvgCovGap across all dataset-model pairs.
> > >
> > > &nbsp;
> > >
> > > ### Runtime
> > >
> > > Thank you for the clarification. Following your comment, we will not include in the revised version the statement: "its runtime is negligible compared to the offline training of DNNs", which relates to the comparison between the runtime of the proposed guidelines and the model training. That said, we still wish to highlight the efficiency of our approach. For example, on our most demanding dataset-model pair, ImageNet-ViT, the entire procedure took under 6 minutes, and is done \textbf{offline} during the calibration phase.
> > >
> > > &nbsp;
> > >
> > > ### Error bars
> > >
> > > Thank you for raising this point. In the following **link**: https://postimg.cc/bSdJqgtg , we present the main metrics (Figure 1), including the AvgCovGap metric with error bars ($\pm$ standard deviation). The variability is minor compared to the trends in the mean metrics; thus, **our interpretations of these figures are unaffected**.
> > > In the revised version, we will include error bars in all presented tables and figures.
> > >
> > > &nbsp;
> > >
> > > We hope our revisions and clarifications satisfy your concerns and contribute to a more positive evaluation.
> > >
> > > Best,
> > >
> > > The Authors

---

### Decision · Program_Chairs · 2025-05-01

**Decision:**

Accept (poster)

**Comment:**

The paper presents the first systematic study of how temperature scaling interacts with adaptive conformal prediction, combining an extensive empirical survey with a supporting mathematical analysis that yields practical tuning guidelines. During the discussion, the authors corrected proof slips, supplied extra examples for an assumption, committed to add average class‑coverage metrics, error bars and runtime figures, and agreed to cite concurrently posted related work. There are a significant number of points that would still need attention in the camera‑ready version, such as: notation and metric definitions require clearer presentation, the technical assumption on sample similarity should be explicitly scoped and backed by the promised additional evidence, and the new evaluation material must be incorporated as agreed. With all of these changes the paper would provide a useful contribution. In consultation with the SAC we recommend weak accept.